



# How does initial soil moisture influence the hydrological response? A case study from southern France

Magdalena Uber[1,3*], Jean-Pierre Vandervaere[1], Isabella Zin[1], Isabelle Braud[2], Maik Heistermann[3], Cédric Legoût[1], Gilles Molinié[1], Guillaume Nord[1]

[1]Univ. Grenoble Alpes, CNRS, IRD, Grenoble-INP, IGE Grenoble, 38000, France
[2]Irstea, UR RiverLy, Lyon-Villeurbanne Centre, Villeurbanne, 69625, France
[3]Institute of Earth and Environmental Science, University of Potsdam, Potsdam, 14476, Germany
*Now at: Univ. Grenoble Alpes, CNRS, IRD, Grenoble-INP, IGE Grenoble, 38000, France

*Correspondence to*: Magdalena Uber (magdalena.uber@univ-grenoble-alpes.fr)

**Abstract.** The Cévennes-Vivarais region in southern France is prone to high intensity and long lasting rainfalls that can lead to flash floods which are one of the most hazardous natural risks in Europe. The results of numerous studies show that besides rainfall depth and intensity and catchment characteristics such as topography, geology, land use and hydraulic routing, the catchment's initial soil moisture also impacts the hydrological response to rain events. The aim of this paper is to analyze the relationship between catchment mean initial soil moisture $\tilde{\theta}_{\mathrm{ini}}$ and the hydrological response that is quantified using the event-based runoff coefficient $\phi_{\mathrm{ev}}$ in the two nested catchments of the Gazel (3.4 km$^2$) and the Claduègne (43 km$^2$). To this end, two research questions are addressed: (1) How heterogeneous are soil moisture patterns in space and time and do they correlate with land use? (2) How does soil moisture at the event onset affect the hydrological response?

The estimation of soil moisture at catchment scale is hindered by high spatial and temporal variability. A sampling setup including 45 permanently installed frequency domain reflectancy probes that continuously measure volumetric soil moisture at three depths is applied. Additionally, on-alert measurements of soil moisture in the topsoil at $\approx$ 10 locations in each one of 11 plots are conducted. Thus, catchment mean soil moisture can be confidently assessed with a standard error of the mean of $\leq$1.7 vol% over a wide range of soil moisture conditions.

$\phi_{\mathrm{ev}}$ is calculated from high-resolution discharge and precipitation data for several rain events with a cumulative precipitation $P_{\mathrm{cum}}$ ranging from less than 5 mm to more than 80 mm. Because of the high uncertainty of $\phi_{\mathrm{ev}}$ associated to the hydrograph separation method, $\phi_{\mathrm{ev}}$ is calculated with several methods, including graphical methods, digital filters and a tracer based method. The results indicate that the hydrological response depends on $\tilde{\theta}_{\mathrm{ini}}$: the seasonal as well as the within-event discharge dynamics follow that of soil moisture. During dry conditions $\phi_{\mathrm{ev}}$ is consistently close to zero, even for events with high and intense precipitation. Above a threshold of $\tilde{\theta}_{\mathrm{ini}}$ = 34 vol% $\phi_{\mathrm{ev}}$ can reach values up to 0.99 but there is a high scatter. Some variability can be explained with a weak correlation of $\phi_{\mathrm{ev}}$ with $P_{\mathrm{cum}}$ and rain intensity, but a considerable part of the variability remains unexplained.



It is concluded that threshold-based methods can be helpful to prevent overestimation of the hydrological response during dry catchment conditions. The impact of soil moisture on the hydrological response during wet catchment conditions, however, is still insufficiently understood and cannot be generalized based on the present results.

# 1 Introduction

The Cévennes-Vivarais region in southern France is prone to intense rainfall that can lead to the occurrence of flash floods in catchments of various scales ranging from small headwater catchments to ones of several thousand km$^2$ (Boudevillain et al., 2011; Braud et al., 2014). Flash floods are sudden floods with high peak discharges of $> 0.5$ m$^3$ s$^{-1}$ km$^{-2}$ (Gaume et al., 2009) and a short rise of the hydrograph, i.e. a time to peak of few hours for catchments with a size of up to 100 km$^2$ and less than 24 h for catchments of up to 1000 km$^2$ (Braud et al., 2014). They are one of the most destructive natural hazards in Europe,

both in terms of number of fatalities and economic damage (Gaume et al., 2009). Unlike lowland floods they often result in losses of life, striking examples being the October 2015 flash flood of the Brague river that hit the French Riviera and the 2002 flash flood of the Gard river with 23 deaths and an estimated direct tangible damage of 1.2 billion Euro (Huet et al., 2003).

Despite the recognition of their high damage potential, the hydrological processes leading to the generation of flash floods

are still insufficiently understood at a scale that is important for prediction and management (Gaume et al., 2009; Braud et al., 2014). Three main problems are recognized to hinder flash flood prediction (Creutin and Borga, 2003): the change-of-scale problem (Blöschl and Sivapalan, 1995) which is especially relevant for capturing the highly heterogenous rainfall fields causing flash floods (Borga, 2002; Creutin and Borga, 2003; Delrieu et al., 2014); the predictions-in-ungauged-basins (PUB) problem (Sivapalan, 2003) and the problem of knowing the soil water retention capacity. Soil moisture is known to

govern overland flow generation (Zehe and Sivapalan, 2008). As it controls threshold behavior, it implies qualitative changes of hydrological processes and the hydrologic system's response to rain events (Zehe and Sivapalan, 2008). Because they are spatially and temporally distinct events, flash floods are difficult to capture with the operational hydro-meteorological measuring systems that are not dense enough to document discrete, rapidly occurring flood events at small scales. Thus, they remain a poorly documented phenomenon (Creutin and Borga, 2003; Borga et al., 2008; Gaume et al.,

2009; Braud et al., 2014). The lack of high-resolution data as well as the variety of catchment characteristics that influence the occurrence of flash floods and the high degree of non-linearity in the hydrological response of catchments hinder the predictability of flash floods (Braud et al., 2014). This has motivated the installation of several measurement networks in first-order catchments especially in the USA and Australia and - at the mesoscale and in a Mediterranean context - the FloodScale project in the Cévennes-Vivarais region (Braud et al., 2014, Nord et al., 2017).

Flash floods are usually associated with intense rainfall of $> 100$ mm in a few hours or long lasting rainfall ($\approx 24$ h) with moderate intensities (Braud et al., 2014) often generated by mesoscale convective systems and / or orographic precipitation (Marchi et al., 2010; Molinié et al., 2012; Panziera et al., 2015). However, the hydrological response to rain events varies



greatly between catchments and between events. It can be quantified using the event-based runoff coefficient $\phi_{ev}$, i.e. the ratio of event runoff volume to total event rainfall volume. The major drawback of this quantity is the lack of standard procedures for obtaining event runoff volumes and for defining the beginning and end of an event, which impedes comparisons between studies (Blume et al., 2007). Yet, event-based runoff coefficients of flash-flood events have been

found to differ substantially, spanning nearly the full range of values from zero to one, with a high positive skewness in their frequency distribution (Merz et al., 2006; Blume et al., 2007; Norbiato et al., 2008; Merz and Blöschl, 2009; Marchi et al., 2010). They were shown to differ considerably between regions (Marchi et al., 2010) and flood types (Merz et al., 2006), to increase with mean annual precipitation and event rainfall depth (Merz et al., 2006; Norbiato et al., 2009) and to depend on rain intensity, soil types and antecedent soil moisture conditions (Wood et al., 1990; Marchi et al., 2010; Huza et al., 2014).

Furthermore, a multitude of catchment characteristics also determine runoff generation and concentration, namely topography, geology, hydraulic routing and geomorphological controls (Braud et al., 2014).

Initial soil moisture $\theta_{ini}$, i.e. the soil water content at the onset of a rain event, is a crucial factor that influences the water storage capacity of the catchment as well as soil hydraulic properties and thus the hydrological response to rainfall events. It controls the soil moisture deficit, consequently, in the interplay with rainfall forcing, it determines whether soil saturation

and saturation excess overland flow (Dunne and Black, 1970) occur during a rain event or not. Moreover, soil moisture controls the unsaturated hydraulic conductivity and thus the occurrence of infiltration excess overland flow (Horton, 1933).

Several studies consider the impact of initial soil moisture on the hydrological response of catchments on heavy rain events. Seasonality in the occurrence of flash floods (Gaume et al., 2009) and discharge magnitudes (Borga et al., 2007) have been attributed to initial soil moisture conditions. The dependence of catchment response to initial soil moisture is also observed

by Marchi et al. (2010) in a dataset comprising data for 25 flood events in 60 basins across Europe and on this study's site by Huza et al. (2014). This relationship is characterized by high non-linearity and threshold effects (Zehe et al., 2005; Huza et al., 2014). There is no consent on the importance of initial soil moisture during extreme events. Wood et al. (1990) conclude that catchment characteristics are important only for flood events with a low return period (up to ca. 10 y) whereas rainfall characteristics dominate those with a higher return period. On the other hand this finding is rejected in analyses of historic

flash floods (Gaume et al., 2004; Borga et al., 2007) or flash flood data bases (Marchi et al., 2010) whose authors conclude that soil moisture plays an important role in the hydrological response, also under extreme conditions.

The aim of this study is to assess how soil moisture controls the hydrological response in a flash-flood prone area in southern France. The study is conducted in the two nested catchments of the Claduègne (43 km2) and Gazel (3.4 km2), Ardèche, France. Thanks to an exceptionally good data basis, it is possible to obtain reliable estimates of the two catchments' initial

soil moisture states for several rain events and to quantify the hydrological response with the event-based runoff coefficient. To this end the spatio-temporal heterogeneity is assessed to obtain reliable estimates for mean initial soil moisture at the catchment scale as well as its uncertainty. It is examined whether soil moisture correlates with land use as this finding could improve the interpolation of point measurements to catchment means. Land use was chosen for this analysis as it controls soil moisture and its variability via interception and evapotranspiration (Reynolds, 1970; Grayson et al., 1997; Garcia-



Estringana et al., 2013) and has a crucial influence on soil hydraulic properties, in particular soil hydraulic conductivity (Gonzales-Soza et al., 2010; Jarvis et al., 2013). Moreover, it is easy to assess in the field or from remote sensing data. In the study site land use correlates well with soil texture in that the vineyards are found on finer textured soils than the other land use classes.

Other studies results that suggest a dependence of $\phi_{ev}$ on initial soil moisture (Merz et al., 2006; Blume et al., 2007; Merz and Blöschl, 2009; Norbiato et al., 2009). However, most of these studies use indirect information such as antecedent precipitation indices, initial baseflow, continuous soil moisture accounting models, the ratio of actual evaporation to precipitation or remote sensing data. At this studies site the impact of $\theta_{ini}$ on $\phi_{ev}$ was already considered by Huza et al. (2014). However, these authors used soil moisture data obtained from ASCAT satellite data which is fitted to in-situ

measurements of topsoil moisture that were conducted on grasslands only. They considered five rain events only, so this relation could not be quantified unambiguously. Thus, this study's objective is to analyze the relation between $\phi_{ev}$ and $\theta_{ini}$ when both are obtained from a comprehensive, high resolution data set. We aim to answer two research questions: (1) How heterogeneous are soil moisture patterns in space and time and do they correlate with land use? And (2) how does soil moisture at the event onset affect the hydrological response?

## 15 2 Methods

### 2.1 Study site

For this study two nested sub-catchments of the Ardèche river in the Cévennes-Vivarais region of southern France are considered: the catchments of the intermittent Gazel stream and the perennial Claduègne river with an area of 3.4 and 43 km$^2$ respectively (Fig. 1).

Both catchments can be clearly divided into two distinct geologies: the northern part is constituted by the Coiron basaltic plateau that is bounded by a steep cliff of basaltic columns in the south, whereas the southern part of both catchments is a landscape of piedmont hills underlain by sedimentary limestone lithology (Nord et al., 2017). The basaltic plateau covers 51 % of the Claduègne catchment whereas its fraction of the Gazel catchment is only 23 %. Thus, the northern part is dominated by silty and stony soils on pebble deposit of basaltic component, while the soils in the southern part are

predominantly rendzinas or other clay-stony soils, cultivated soils of loam and clay-loam and in the south of the Claduègne catchment lithosols and regosols (Nord et al., 2017). The terrain is hilly, ranging from about 820 m above sea level (asl) to 230 m asl at the outlet (650–260 m asl for the Gazel catchment) having a mean slope of about 20 %. The area is characterized by extensive agriculture and natural vegetation. Hence, the main land use / land cover types are grasslands, pastures, vineyards, forests and Mediterranean open woodlands. The vineyards are predominantly found on the finer textured

soils in the southern part of the Claduègne catchment while the other land use types are found throughout the catchments. The average annual precipitation at Le Pradel at the outlet of the Gazel catchment is 1030 mm (Huza et al. (2014), original data: daily rain gauge data for 1958–2000 from Méteo-France). For further details see Nord et al. (2017).





## 2.2 Data availability

As part of the HyMex (Hydrological Cycle in the Mediterranean Experiment, Ducrocq et al. (2014)) and FloodScale (Braud et al., 2014) projects and the Cévennes-Vivarais Mediterranean Hydrometeorological Observatory (OHM-CV, Boudevillain et al. (2011)), the area is exceptionally well monitored, thus high-resolution spatio-temporal data on rainfall, discharge and
soil moisture is available. The data used for this study was published in Nord et al. (2017) and the link to download the data can be found at the publishers website: https://www.earth-syst-sci-data.net/9/221/2017/essd-9-221-2017-assets.html.

**Soil moisture $\theta$**: two different sets of soil moisture data are available: continuous measurements and on-alert measurements. Soil moisture is continuously measured with 45 fixed soil moisture probes at nine plots (two vineyards, one fallow, six grasslands) within the Claduègne catchment since June 2013. Six of the plots are located in the piedmont hills and three on
the basaltic plateau (Fig. 1). At each plot, five frequency domain reflectometry (FDR) probes (Decagon 10HS soil moisture sensors) are installed at different depths: 10 cm ($n = 2$), 20–25 cm ($n = 2$) and in the subsoil (33–50 cm, $n = 1$). The temporal resolution is 15–20 min (Nord et al., 2017). The sensors in the vineyards were installed between two vine plants in a row. The accuracy and the range of the probe as provided by the manufacturer are ±3 vol% and 0–57 vol%. The data is available from June 2006 – November 2014 in the dataset doi: 10.17178/OHMCV.SMO.CLA.13-14.1 presented in Nord et al. (2017).
In addition, following forecasts of heavy rain events on-alert measurements of soil moisture in 0–5 cm depth were conducted at 11 plots within the Gazel catchment with a hand-held FDR soil moisture sensor (Delta-T SM200). The accuracy and the range of the probe are ±3 vol% and 0–50 vol%. The plots comprised four vineyards, five grasslands, one fallow and one cultivated field. The sampling sites were selected for reasons of accessibility, congruence with other measurements conducted during the FloodScale project and representativeness of the catchments' landscapes. All on-alert measurements
were conducted in about 1 h at ≈ 10 randomly chosen measurement points within each plot. The distance between the measurement points was at least 1 m to ensure spatial independence (Huza et al., 2014). In the vineyards the measurements were conducted in between the rows of vine plants. On-alert measurements were done before and after 11 heavy rain events during the special observation periods of the HyMex Project in autumn (September–December) of the years 2012–2015. The dataset is found in the supplementary material of this article (S1).
**Precipitation $P$**: rainfall data was obtained from the HPiconet rain gauge network at a resolution of 1 min. The network consists of 19 tipping bucket rainfall gauges with a sampling surface of 1000 cm$^2$ and a resolution of 0.2 mm out of which 12 are located in the Claduègne catchment or its close vicinity (Nord et al. (2017), DOI: 10.17178/OHMCV.RTS.AUZ.10-14.1 Fig. 1).

**Discharge $Q$**: water level is continuously measured at the outlets of the two catchments with water level gauges at 2 min
resolution (Gazel) or 10 min resolution (Claduègne) respectively (Nord et al., 2017). The water level is converted to discharge with a stage-discharge relationship established using the BaRatin framework (Le Coz et al., 2014) that also gives the uncertainty of the rating curve that is quantified as the 90 % confidence interval of discharge. The rating curve is based on numerous discharge measurements in 2012–2014 (Nord et al., 2017).





**Additional data**: spatial data used for this study include a digital elevation model with a resolution of 5 m (DOI: 10.6096/MISTRALS-HyMeX.1389) and the Ardèche soil data base by the French National Institute for Agricultural Research, Bureau de Recherches Géologiques et Minières and the French Department of Agriculture (DOI: 10.6096/MISTRALS-HyMeX.1385, Nord et al., 2017). Furthermore, a 0.5 m resolution land use map of the Claduègne

catchment based on quickbird satellite images is available (DOI: 10.14768/mistrals-hymex.1381, Andrieu, 2015). Data on soil properties such as porosity, texture and saturated hydraulic conductivity was obtained during a measurement campaign in 2012 by Braud and Vandervaere (2015). Electrical conductivity (*EC*) of stream flow is continuously measured at the outlets of both catchments and measurements of *EC* of overland flow from two runoff and erosion plots in a vineyard in the south of the Gazel catchment are available (Cea et al., 2015).

**2.3 Precipitation data processing**

The catchment mean hyetographs for both catchments are calculated from the HPiconet rain gauge data with the method of Thiessen polygons. Rain events are separated by using a threshold of 12 h without precipitation being recorded at any rain gauge. The onset of an event was defined as the first time rain occurred after a dry period of at least 12 h, the end as the last time with rain being recorded by at least one rain gauge before the next dry period. The threshold of 12 h provides a good

compromise between having a high number of events and excluding to separate events that are not independent from each other. Averaged catchment rainfall is then summed over the whole period of the rain event to calculate cumulative event precipitation $P_{cum}$. Furthermore, mean rain intensity $I_\mu$ over the whole event as well as maximum rain intensity $I_{max}$ at 2, 10, 20, 30 and 60 minutes are calculated using the averaged catchment rainfall.

**2.4 Soil moisture analysis**

From both data sets (continuous and on-alert measurements) plot and catchment mean values are calculated for the initial and final state of each rain event. From the continuous data, mean values are calculated for all three depths and the profile mean value is calculated.

$\theta(x_{i,j}, t_{ev})$ refers to a spatially and temporally discrete on-alert soil moisture measurement, with the subscript i denoting the index of the $n_i$ (usually ten) measurements within the plot, j denoting the plot and ev the event and the state (initial or final).

Plot mean soil moisture $\overline{\theta}_j(t_{ev})$ of the $n_i$ local measurements were calculated for all plots as well as land use class means $\overline{\overline{\theta}}_{lu}(t_{ev})$ of $n_{j,lu}$ $n_{j_{lu}}$ plots belonging to the same of the four land use classes grassland, vineyard, fallow and cultivated field ($c_{lu} = \{g,v,f,c\}$):

$$\overline{\theta}_j(t_{ev}) = \frac{1}{n_i} \sum_{i=1}^{n_i} \theta\left(x_{i,j}, t_{ev}\right) \#(1)$$





$$\bar{\bar{\theta}}_{\text{lu}}(t_{\text{ev}}) = \frac{1}{n_{\text{j}_{\text{lu}}}} \sum_{\text{j}_{\text{lu}}=1}^{n_{\text{j}_{\text{lu}}}} \bar{\theta}_{\text{j}}(t_{\text{ev}}); \qquad \text{j}_{\text{lu}} \in \text{lu}; \text{lu} \in c_{\text{lu}} \quad \#(2)$$

with $n_{\text{j}_{\text{lu}}}$ = number of plots in the respective land use class.

Catchment mean values $\bar{\bar{\theta}}_{\text{ev}}$ are computed as a mean of the different land use classes weighted with the number of measurements per land use class:

$$\bar{\bar{\bar{\theta}}}_{\text{ev}} = \frac{1}{n_{\text{p}}} \sum_{\text{lu}}^{n_{c_{\text{lu}}}} \bar{\bar{\theta}}_{\text{lu}} \cdot n_{\text{j}_{\text{lu}}}; \qquad \text{lu} \in c_{\text{lu}} \quad \#(3)$$

with $n_{c_{\text{lu}}}$ = number of land use classes ($n_{c_{\text{lu}}} = 4$) and $n_{\text{p}}$ = total number of plots ($n_{\text{p}} = \sum n_{\text{j}_{\text{lu}}} = 11$).

Plot means and catchment means for the continuously measured data are computed in the same way for all three layers $l$ (therefore denoted $\bar{\bar{\bar{\theta}}}_{\text{ev},l}$) with the exception, that the plot mean is obtained by averaging not only the probes installed at the same depth and the same location, but also several measurements in a dry period of two hours before the onset or after the end of the rain event in order to diminish noise.

The profile mean $\tilde{\theta}_{\text{ev}}$ over the three layers ($n_l$ = number of layers) is calculated with the following equation. The thicknesses

$m_l$ of the layers are assumed to be 175, 150 and 275 mm respectively, hence, the topmost 60 cm are considered:

$$\tilde{\theta}_{\text{ev}} = \frac{\sum_{l=1}^{n_l} \bar{\bar{\bar{\theta}}}_{\text{ev},l} \cdot m_l}{\sum_{l=1}^{n_l} m_l} \quad \#(4)$$

Finally, for all events the soil moisture storage change $\Delta S$ [mm/ev] in the upper 60 cm is computed from the difference between initial ($\bar{\bar{\bar{\theta}}}_{\text{ini},l}$ [vol%]) and final soil moisture ($\bar{\bar{\bar{\theta}}}_{\text{fin},l}$ [vol%]) that is converted to [vol/vol] via division by 100:

$$\Delta S = \sum_{l=1}^{n_l} \frac{1}{100} \left( \bar{\bar{\bar{\theta}}}_{\text{fin},l} - \bar{\bar{\bar{\theta}}}_{\text{ini},l} \right) \cdot m_l \quad \#(5)$$

For all plots and all events, the frequency distributions of the on-alert soil moisture measurements are tested for normality with a Shapiro-Wilk test at a significance level of $\alpha = 0.05$. It is also assessed whether significant differences between the

four land use classes exist by performing a visual inspection of boxplots or histograms and Student t-tests. Moreover, standard deviations as measures of spatial variability at different scales are calculated: the inner-plot standard deviation $\sigma_{\text{j}}^{\text{inner}}$ and the inter-plot standard deviation for the land use classes grassland and vineyard ($\sigma_{\text{lu}}^{\text{inter}}$; $n_{\text{j} \in \text{lu}} > 1$) and the whole catchment ($\sigma_{\text{cat.}}^{\text{inter}}$) are calculated for each event:





$$\sigma_j^{inner}(t_{ev}) = \sqrt{\frac{1}{n_i - 1} \sum_{i=1}^{n_i} (\theta(x_{i,j}, t_{ev}) - \overline{\theta}_j(t_{ev}))^2 \#\#\#\#} \#(6)$$

$$\sigma_{lu}^{inter}(t_{ev}) = \sqrt{\frac{1}{n_{j_{lu}} - 1} \sum_{j_{lu}=1}^{n_{j_{lu}}} (\overline{\theta}_j(t_{ev}) - \overline{\overline{\theta}}_{lu}(t_{ev}))^2}; \qquad j_{lu} \in lu; lu \in c_{lu} \#(7)$$

$$\sigma_{cat.}^{inter}(t_{ev}) = \sqrt{\frac{1}{n_p - 1} \sum_{j=1}^{n_p} (\overline{\theta}_j(t_{ev}) - \overline{\overline{\overline{\theta}}}(t_{ev}))^2} \#(8)$$

Furthermore, the between-land use standard deviation $\sigma^{betw}$ is computed:

$$\sigma^{betw}(t_{ev}) = \sqrt{\frac{1}{n_{c_{lu}} - 1} \sum_{lu}^{n_{c_{lu}}} (\overline{\overline{\theta}}_{lu}(t_{ev}) - \overline{\overline{\overline{\theta}}}_{ev})^2} \#(9)$$

As an estimate of the uncertainty of the calculated plot and catchment mean values, the standard error of the plot mean $SEM_j^{inner}$ and the one of the catchment mean $SEM_{cat.}^{inter}$ are calculated with Eq. (10) and Eq. (11). It should be noted that the latter is calculated from the on-alert measurements in the topsoil as well as from the continuous measurements over the soil profile, the former only from the on-alert measurements. The $SEM$ is used as a measure of the confidence that the sample mean corresponds to the universal mean; it increases with the standard deviation and decreases with the number of sampling points:

$$SEM_j^{inner}(t_{ev}) = \frac{\sigma_j^{inner}(t_{ev})}{\sqrt{n_i}} \#(10)$$

$$SEM_{cat.}^{inter}(t_{ev}) = \frac{\sigma^{betw}(t_{ev})}{\sqrt{n_p}} \#(11)$$

Moreover, it is assessed whether temporal stability, i.e. consistency of soil moisture patterns at the catchment scale at different times of measuring (Vachaud et al., 1985), as reported by Huza et al. (2014) for six grassland plots in the Gazel catchment, is also found in the present on-alert data set: the relative spatial difference $\delta_{j,ev}$ of each plot corresponds to the relative difference between the plot mean and the catchment mean (Eq. 12); its temporal mean $\overline{\delta}_j$ is calculated with Eq. (13):

$$\delta_{j,ev} = \frac{\overline{\theta}_j(t_{ev}) - \overline{\overline{\overline{\theta}}}_{ev}}{\overline{\overline{\overline{\theta}}}_{ev}} \#(12)$$





$$\overline{\delta_{\mathrm{j}}} = \frac{1}{n_{\mathrm{ev}}} \sum_{\mathrm{ev}=1}^{n_{\mathrm{ev}}} \delta_{\mathrm{j,ev}} \quad \#(13)$$

The plot with the smallest $\delta_{\mathrm{j,ev}}$ is the one that agrees best with the catchment mean on a given time of measurement. The temporal variability of the spatial difference $\sigma_{\delta_{\mathrm{j}}}$ serves as an auxillary variable to assess whether this behavior is stable in time:

$$\sigma_{\delta_{\mathrm{j}}} = \sqrt{\frac{1}{n_{\mathrm{ev}}-1} \sum_{\mathrm{ev}=1}^{n_{\mathrm{ev}}} (\delta_{\mathrm{j,ev}} - \overline{\delta_{\mathrm{j}}})^2} \quad \#(14)$$

The temporal dynamics of soil moisture during the autumn season and during single events that are captured with the continuous soil moisture measurements are analyzed and compared with the hyetographs and hydrographs. Furthermore, crosscorrelations are calculated for all continuously measuring sensors and all depths with the R function `ccf` (Gilbert and Plummer, 2016). For each plot the maxima of the crosscorrelation functions $L_{\mathrm{CC_{max}}}$ between rainfall and the sensors in 10 cm depth, between the ones in 10 and 25 cm and the ones in 25 and 40 cm are calculated.

## 2.5 Hydrological response

### 2.5.1 Event based runoff coefficients

In order to quantify the hydrological response of the catchment to different rainfall events, the dimensionless event-based runoff coefficient $\phi_{\mathrm{ev}}$ is calculated for all events:

$$\phi_{\mathrm{ev}} = \frac{Q_{\mathrm{ev,cum}}}{P_{\mathrm{cum}}} \quad \#(15)$$

$P_{\mathrm{cum}}$ is calculated for each event as described in Sect. 2.3. To obtain cumulative event discharge $Q_{\mathrm{ev,cum}}$, the time series of stream discharge $Q_{\mathrm{tot}}$ has to be separated into baseflow $Q_{\mathrm{b}}$ and event flow $Q_{\mathrm{ev}}$. $Q_{\mathrm{ev}}$ is defined here to be the fast responding part of discharge that occurs during or directly after the rain event. It usually encompasses surface runoff or overland flow and fast subsurface flow. $Q_{\mathrm{b}}$ on the other hand is the slow responding part of discharge that lasts long after the rain event and feeds the stream between rain events. To obtain $Q_{\mathrm{ev,cum}}$, $Q_{\mathrm{ev}}$ is summed up over the whole period of the event. The onset of a discharge event is defined as the first increase of discharge in response to a rain event. Defining the end of event discharge is more complicated and depends on the hydrograph separation method (Blume et al., 2007). Usually, the end of event flow is defined as the moment when $Q_{\mathrm{b}}$ equals $Q_{\mathrm{tot}}$, but for some events the onset of a second event impedes this procedure which causes errors. Taking into consideration that there is no standard method for hydrograph separation and that results obtained with different methods can differ substantially (Blume et al., 2007), seven different hydrograph separation techniques that are described in the following section are applied and compared (Fig. 2). The uncertainty of $Q_{\mathrm{tot}}$ associated with the stage-





discharge relation can be important especially for high-flow conditions. This was taken into account by calculating $\phi_{\text{ev}}$ with the upper and the lower limit of the 90 % confidence interval of discharge obtained with the BaRatin framework.

Because several of the assumptions underlying standard regression analysis methods (normal distribution of dependent and independent variables, only the dependent variable is subject to error, homoscedasticity) are not met, it is not attempted to

set-up linear or non-linear regression models that predict $\phi_{\text{ev}}$ from possible explanatory variables such as initial soil moisture, initial baseflow, rainfall depth or rain intensity. The relation of these variables and $\phi_{\text{ev}}$ is considered nonetheless.

### 2.5.2 Hydrograph separation

**Straight line method.** The straight line method (SL) is a simple, graphical method where baseflow during the event is interpolated by connecting the point in the event hydrograph at which discharge first increases as a response to the rain event

with the first point on the falling limb of the hydrograph with the same discharge value. As this condition is often never met, the end of event flow is often determined by the onset of the next event or discharge below a threshold.

**Constant-k method.** The CK-method proposed by Blume et al. (2007) is based on the assumption that baseflow recession behaves similar to the outflow of a linear storage. Thus, baseflow at time step $t$ can be described as exponential recession with the recession parameter $k$ and initial flow $Q_0$:

$$Q_{\text{b}}(t) = Q_0\, \mathrm{e}^{-k\,t} \#(16)$$

$k$ is calculated at each time step by differentiating eq. 16 and division by $Q_{\text{b}}(t)$:

$$k = -\frac{\mathrm{d}Q}{\mathrm{d}t} \frac{1}{Q_{\text{b}}(t)} \#(17)$$

Event flow is assumed to terminate at time step $t_{\text{e}}$ which is defined as the end of event runoff, once $k$ becomes approximately constant. Baseflow is assumed to be equal to the discharge before the onset of event flow up to $t_{\text{e}}$ when it equals $Q_{\text{tot}}$.

**Electrical conductivity method.** Hydrograph separation is also conducted based on electrical conductivity ($EC$) which serves as a natural tracer (Miller et al., 2014; Pellerin et al., 2008). The method relies on the assumption that stream flow $Q_{\text{tot}}$

with electrical conductivity $EC_{\text{tot}}$ is a mixture of subsurface flow $Q_{\text{sb}}$ and surface flow $Q_{\text{s}}$, which have significantly different $EC$ signals $EC_{\text{sb}}$ and $EC_{\text{s}}$ (Nakamura, 1971):

$$Q_{\text{tot}} = Q_{\text{sb}} + Q_{\text{s}} \#(18)$$

$$Q_{\text{tot}} \cdot EC_{\text{tot}} = Q_{\text{sb}} \cdot EC_{\text{sb}} + Q_{\text{s}} \cdot EC_{\text{s}} \#(19)$$

Thus, with given values for $EC_{\text{sb}}$ (interpolated $EC$ between values before the onset and after the end of event discharge) and $EC_{\text{s}}$ (measured in overland flow collected at the outlet of four erosion plots representative of the signature of the rainfall flowing at the surface of soils developed on sedimentary rocks), a time series of $Q_{\text{sb}}$ can be calculated. Furthermore, the

maximum signal of $EC$ is calculated for each event as $\Delta EC_{\text{max}} = \max(|EC_{\text{sb}} - EC_{\text{tot}}|)$. As no $EC_{\text{s}}$ values were available for overland flow occurring on soils developed on basalts, covering half of the Claduègne catchment, the method could only be applied to the Gazel catchment, where the proportion of basaltic geology to total catchment surface is much smaller.





It should be noted that this method considers only the surficial part of event flow and is not able to separate the fast responding subsurficial flow occurring in the unsaturated zone. Thus, event flow is likely to be underestimated, especially in conditions when the latter plays an important role.

**Recursive digital filter.** The RDF method proposed by Lyne and Hollick (1979) uses a signal analysis low-pass filter to separate high-frequency event flow signals from low frequency baseflow signals:

$$f(t) = a\,f(t-1) + \frac{1+a}{2}\big(Q_{\text{tot}}(t) - Q_{\text{tot}}(t-1)\big) \#(19)$$

where $f(t)$ is filtered event flow at time $t$, $a$ is a filter parameter that is usually in the range of $0.00 < a < 0.95$ (Nathan and McMahon, 1990) and $Q_{tot}(t)$ is original stream flow at time $t$. The data is passed to the filter several times, forwards and backwards. Recommendations for the number of passes vary depending on the time increment of the discharge series (Ladson et al., 2013). The method is implemented in the R function `BaseflowSeparation` of the package `EcoHydRology` (Archibald, 2014).

**Hysep filters (HS1 – HS3).** Three further filtering approaches are implemented in the Unites Stated Geological Survey's (USGS) HySep program (Sloto and Crouse, 1996). It is based on finding minima in the discharge time series. The minima are determined either within fixed (HS1) or sliding (HS2) intervals or with a local minima algorithm (HS3). The interval width is adjusted according to Gonzales et al. (2009). It is applied using the R code of the USGS (2015).

# 3 Results

## 3.1 Spatial variability of soil moisture

### 3.1.1 Soil moisture at the plot scale

The probability density function (pdf) of the on-alert measurements of soil moisture of the topsoil measured within one plot at a discrete moment in time $\theta(x_{\text{i,j}}, t_{\text{ev}})$ can usually be described with a normal distribution (230 out of 243 within-plot measurements; Shapiro-Wilk test with $\alpha = 0.05$, Fig. 3a). In the present data set, normal distribution is found during dry, medium and wet conditions. Out of the 23 measurements conducted during dry conditions ($\overline{\theta}_{\text{j}}(t_{\text{ev}}) < 19$ vol%, the 10 % quantile) and the 23 measurements during wet conditions ($\overline{\theta}_{\text{j}}(t_{\text{ev}}) > 37$ vol%, the 90 % quantile) only one and zero respectively data sets are not normally distributed.

The variability of soil moisture at the plot scale, determined from the on-alert measurements in the topsoil, is very high: the median range between the highest and the lowest measurement in one plot is 7.8 vol% but maximum values can get up to > 30 vol%. The mean of the inner-plot standard deviation is 2.7 vol%. Values range from 1–8 vol% with no significant difference between the land-use classes. There is no significant correlation between plot means and standard deviations (Fig. 3c). The inner-plot standard deviation in the deeper layers, determined with the continuously measuring probes, cannot be confidently assessed because of the low number of probes installed in each plot at the same depth. However, the difference



of two sensors installed at the same depth indicate, that the variability is in the order of the one derived from the on-alert measurements (Table 1).

The mean $SEM_j^{inner}(t_{ev})$ is 0.8 vol% with only three out of 228 data sets exceeding 2.0 vol%. Thus, the confidence that the population plot mean lies within the sample mean $\overline{\theta}_j(t_{ev}) \pm 2.0$ vol% is very high.

**3.1.2 Soil moisture at the catchment scale**

The pdf of the on-alert measurements that are conducted at the same time in different plots of the same land use $\theta(x_{i,j}, t_{ev})$ agree either with the normal distribution (29 out of 50 data sets) or show a bi-modal or multi-modal distribution (remaining 21 data sets, Fig. 3b). This is an indication of inter-plot variability within the same land use class. In fact, the mean of the inter-plot variability within one land use class $\sigma_{lu}^{inter}(t_{ev})$ exceeds the mean inner-plot variability of the same land use class $\sigma_{j \in lu}^{inner}(t_{ev})$ almost everywhere, except of in the vineyards at depth 10 cm (Table 1). There is a weak correlation between $\tilde{\theta}(t_{ev})$ and $\sigma_{cat.}^{inter}(t_{ev})$ (Fig. 3d). The mean of the standard error of the Claduègne catchment $SEM_{cat.}^{inter}$ is 1.5 vol%, the maximum is 1.7 vol%. The mean $SEM_{cat.}^{inter}$ of the Gazel catchment is 1.3 vol%, the maximum is 1.7 vol%.

Both the on-alert and the continuous measurements were analyzed for differences between the land use classes. The results of the student t-tests that are conducted for all on-alert measurements to detect differences between soil moisture in the different land use classes are given in Table 2. The null hypothesis (true difference in means is equal to zero) can be rejected under a significance level of α = 0.05 only a limited number of times (upper panel). Furthermore, only the difference between the grasslands and the cultivated field is a systematic one, meaning that for all cases with significant differences the grasslands have a higher soil moisture that the cultivated field (lower panel).

A comparison of the variability between the four land use classes expressed as $\sigma^{betw}(t_{ev})$ to the one within land use classes $\sigma_{lu}^{inter}(t_{ev})$ or within plots $\sigma_j^{inner}(t_{ev})$ also reveals that it is smaller than both other standard deviations (Table 1). The initial and final soil moisture profile of the first major event in 2013 (ev. 27) shows, nonetheless, that there are differences in the profile shape and in the wetting behavior between grasslands and vineyard (Fig. 4). While the grasslands have a nearly homogenous profile before the rain event, the vineyards have a much more pronounced vertical soil moisture profile with higher values in the deeper layers. In response to the rain event, the profile of the grasslands shifts towards higher soil moisture, with similar differences in each depth. In the vineyards, mainly the moisture in the topmost layer increases, whereas soil moisture in the subsoil hardly changes.

Figure 5 shows the relative spatial difference $\delta_{j,ev}$ of all plots for all on-alert measurements conducted from 2012 to 2015. It can be seen that temporal stability is found to some degree. The mean values of some plots are (nearly) consistently below the catchment mean (v4, v3, g5, f1, c1), others above (g4, g3, g1). This is also the case if deviations were related to the land use mean instead of the catchment mean (see e.g. the noticeable difference between g2 and g5 on the one hand and g1, g3 and g4 on the other hand). However, there are also plots with above average soil moisture for a certain period of time and below average soil moisture during other periods, indicated by a change in signs of $\delta_{j,ev}$ between events (v1, v2, g2). The





plots with the lowest mean spatial difference $\overline{\delta_j}$ are v1, v2 and c1 (3, 4 and 8 % respectively). The one with the lowest temporal variability of the relative spatial difference ($\sigma_{\delta_j} = 5$ %) is the cultivated field c1.

## 3.2 Temporal dynamics

Figure 6 shows the evolution of soil moisture in 10, 25 and 40 cm depth in autumn 2013. Due to several large rain events in July and August 2013 (not shown here), soil moisture at the beginning of the season is already ca. 25 vol% at 10 cm depth and ca. 30 vol% in the deeper layers. In the topsoil, however, soil moisture is much lower at the beginning of the season: the catchment mean value of the first measurement conducted on 28 September 2013 is 10.2 vol% (Fig. 7). After the first major rain events, it remains constantly above 30 vol% at 10 cm depth and above 36 vol% in the deeper layers, with maximum values of around 42 vol% reached after major rain events. This value is not exceeded, even after rain events that occur during wet initial conditions.

Temporal variability of soil moisture varies considerably between wet and dry conditions. Soil moisture in all continuously sampled depths increases rapidly as a response to rain events by up to 12.6 vol% in less than 1 d (Fig. 6). Differences between initial and final state in the topsoil can be even larger (maximum difference: 16 vol% in November 2014, Fig. 7). The rapid response is evident from the small lag between the peak of rainfall and the peak of soil moisture. The maximum of the crosscorrelation function ($L_{CC_{max}}$) between hourly rainfall and hourly catchment mean soil moisture at 10 cm depth is at the lag time of 1 h. Mean $L_{CC_{max}}$ between the probes in 10 cm and the probes in 25 cm depths is 0.3 h and the one between 25 and 45 cm is 1.3 h, which indicates a fast percolation to deeper layers. There are differences between grasslands (median $L_{CC_{max}}$ = 0.15 h), vineyards (1.35 h) and the fallow (3.3 h). As these values are in the same order of magnitude as the resolution of the underlying data, their uncertainty is high. Thus, they represent only estimates for the order of magnitude of percolation time.

A few days after the rain events, soil moisture decrease slows down and soil moisture during dry conditions can persist for long time periods as in the first half of December 2013 (Fig. 6).

## 3.3 Event-based runoff coefficients

The event-based runoff coefficients $\phi_{ev}$ calculated for the Gazel catchment with seven hydrograph separation methods for 54 events range from 0 to 0.99 with large differences between the methods and a high positive skewness because of many low values (Fig. 8a). In the Claduègne catchment, $\phi_{ev}$ was only calculated with the recursive digital filter method RDF and the Hysep filter methods, but values still range from 0 to 0.97. The electric conductivity and constant-K methods result in the lowest values for $\phi_{ev}$ while the 3 HySep filters yield considerably larger values than all other methods. Except of the HySep filter methods, the other four methods correlate well with each other (Fig. 8b). The HySep filters correlate very well with each other ($R^2 \geq 0.96$ for all three pairs; not shown here), but to a lesser degree with the other methods (Fig. 8b). For the



following sections, values of $\phi_{ev}$ determined with the RDF method are used for reasons explained in the discussion (Sect. 4.4).

The uncertainty of $\phi_{ev}$ associated with the stage-discharge relation is quantified with the BaRatin method and is shown in Fig. 9 as black vertical error bars. In both catchments this uncertainty is very small for events with low $\phi_{ev}$ while it can get up to 0.28 (difference between $\phi_{ev}$ calculated with the 5 % and the 95 % confidence interval of discharge) for event 40 which is the one with the highest $\phi_{ev}$ and highest discharge in both catchments (Table 3 and Table 4). The uncertainty due to different $\phi_{ev}$ obtained by different hydrograph separation methods is visualized as gray vertical error bars in Fig. 9. It can be very high for any event regardless of the $\phi_{ev}$ and is often due to the discordance of the HySep methods with the other methods. The mean range in $\phi_{ev}$ calculated with different hydrograph separation methods is 0.08, the one of $\phi_{ev}$ calculated with 5 % CI and 95 % CI of discharge is 0.02. The mean standard deviation of $\phi_{ev}$ calculated with different hydrograph separation methods is 0.03, when the HySep methods are excluded it decreases to 0.02. However, these measures are biased by the important positive skewness of the distribution of $\phi_{ev}$.

Factors that are suggested to influence $\phi_{ev}$ include rainfall depth and rain intensity. Figure 9 shows the correlation of the meteorological forcing quantified as cumulative event rainfall depth $P_{cum}$, mean rain intensity $I_\mu$, and maximum 20 min rain intensity $I_{max,20}$ with $\phi_{ev}$. In the present data set there is a weak correlation between $\phi_{ev}$ and $P_{cum}$, $I_\mu$ and $I_{max,20}$ (coefficients of determination $R^2 = 0.24$, 0.17 and 0.27 respectively). The correlations of $\phi_{ev}$ with maximum rain intensity calculated at 2, 10, 30 and 60 min time steps were worse that the one at 20 min. None of these measures can, therefore, explain more than 30 % of the variability of $\phi_{ev}$. Figure 9 shows that events with similar rainfall characteristics (events 30 and 40, similar intensity) can have very different $\phi_{ev}$. Additionally, similar $\phi_{ev}$ are obtained for events with very different rainfall characteristics (events 22 and 39). These striking differences in catchment behavior can partly be attributed to differences in initial soil moisture as shown in the following section.

### 3.4 Soil moisture's impact on runoff generation

The hydrological responses concerning the temporal dynamics of soil moisture and discharge in reaction to rain events of the two catchments vary greatly. In Table 3 and Table 4 the characteristics of all rain events in autumn 2013 that generate event flow at the river gauges of the Gazel and / or Claduègne are given. The hyetographs, hydrographs and time series of catchment mean soil moisture of four of these events with very different behavior are exemplarily shown in Fig. 10: event 27 and event 30 occur at the beginning of the season when soil moisture is still relatively low. Rainfall leads to a considerable increase in soil moisture in all three layers and to a storage change $\Delta S$ that constitutes a notable share of cumulative precipitation. For event 30 on-alert soil moisture measurements are available as well and show a sharp increase from 10.2 vol% before to 25.7 vol% after the event (Fig. 7). The runoff coefficients $\phi_{ev}$ of both events are very low. The within-event temporal dynamic of rainfall, soil moisture and runoff during event 27 is also noteworthy: the discharge peak does not follow the rainfall peak, which is closely followed by the steepest increase in soil moisture, but the second rainfall pulse that occurs when soil moisture is considerably higher than at the beginning of the event. As a response to this much smaller rainfall





impulse, soil moisture rises only slightly. This behavior is also observed during event 40 where the first rainfall impulse leads to a sharp increase in soil moisture and only a small discharge peak, while the second rainfall pulse generates a substantial discharge peak and only a slight increase in soil moisture. Event 40 and event 53 both occur during wet initial soil moisture conditions, but event 53 has a much smaller $\phi_{ev}$ than event 40. During these two events also an inversion of the

vertical soil moisture profile, i.e. temporally higher soil moisture at the topsoil than in deeper layers, can be observed approximately at the time of peak discharge. This inversion is an indicator of Hortonian overland flow.

The large range of $\phi_{ev}$ of events with high $\tilde{\theta}_{ini}$ can also be seen in Fig. 11. While the three events with low $\tilde{\theta}_{ini}$ consistently have very low $\phi_{ev}$, after a threshold of approximately 34 vol%, $\phi_{ev}$ can have a value anywhere between 0 and 1. An examination of Fig. 11 shows that both high $\tilde{\theta}_{ini}$ and high $P_{cum}$ are necessary but not sufficient criteria for high $\phi_{ev}$ and that

the relation between $\tilde{\theta}_{ini}$ and $\phi_{ev}$ is characterized by strong non-linearity and threshold effects. This is observed in both catchments and the threshold value is very similar for the Gazel and the Claduègne catchment. Further analysis of the relation between $\tilde{\theta}_{ini}$ and $\phi_{ev}$ for events with high $P_{cum}$ and high $\tilde{\theta}_{ini}$ is limited because of the low number of events fulfilling these criteria and the uncertainty of both $\tilde{\theta}_{ini}$ and $\phi_{ev}$.

Consideration of single events shows, that $\tilde{\theta}_{ini}$ can partly explain the high scatter in Fig. 9. The contrary behavior of events

30 and 40 can be explained by different initial soil moisture conditions. It can also be hypothesized that the high $\phi_{ev}$ of event 22 despite low $P_{cum}$, $I_\mu$ and $I_{max,20}$ is due to high initial soil moisture. For this event, that started on November 26[th], 2012, only on-alert soil moisture is available and initial topsoil moisture is relatively high at 23.5 vol%. The event ocurred late in the season (26 November 2012, Fig. 7) two weeks after a heavy rain event with $P_{cum}$ = 72.5 mm. Event 39 which has a similar $\phi_{ev}$ but higher $P_{cum}$, $I_\mu$ and $I_{max,20}$, however also occurred during high initial moisture conditions (36 vol%) which indicates

that the relation between $\phi_{ev}$, $\tilde{\theta}_{ini}$ and rainfall characteristics cannot easily be generalized.

The results of hydrograph separation with the electric conductivity method give some insights into runoff generating processes in the Gazel catchment. During many events (e.g. Fig. 12a), there is a very large contribution of subsurface flow to total event flow. This is also reflected in rather low values for $\phi_{ev}$ obtained with the EC method compared to other methods. The subsurface runoff component reacts very quickly during the rising limb of the hydrograph (e.g Fig 12a). Furthermore,

the response of $EC$ differs greatly between events (Fig. 12). A high signal in electric conductivity, $\Delta EC_{max}$ indicates that surface runoff contributes strongly to total stream discharge while a low value indicates a high impact of subsurface stormflow. Values for $\Delta EC_{max}$ for different events are given in Table 3 and Table 4 and shown in Fig. 13. The highest values for $\Delta EC_{max}$ are observed during events 30,39 and 40 suggesting a strong contribution of surface runoff for these events. In fact, for event 40, overland flow was observed during a field visit on vast areas in the north of the Claduègne catchment (see

pictures in the supplementary material S2). $\Delta EC_{max}$ generally increases with $P_{cum}$ and peak discharge $Q_p$, but there is no correlation with $\tilde{\theta}_{ini}$ ($R^2$ = 0.00) and only a moderate one with $I_\mu$ and $I_{max,20}$ ($R^2$ = 0.51 and 0.64 respectively). Figure 13 shows that high values of $\Delta EC_{max}$ are observed for several events with high $\tilde{\theta}_{ini}$ (39, 40, 52). A high value of $\Delta EC_{max}$ is also measured during event 30, with low $\tilde{\theta}_{ini}$. This event has the highest $I_{max,20}$ and considerable cumulative precipitation.





## 4 Discussion

### 4.1 Validity of the soil moisture sampling strategy

The sampling design applied in this project proved to be efficient to assess spatial variability of soil moisture across scales, as well as to document temporal dynamics. The on-alert measurements of soil moisture allow a good estimate of the plot

mean with a low mean $SEM_\mathrm{j}^\mathrm{inner}$ of 0.8 vol% as well as an accurate estimate of the inner-plot variability, quantified as $\sigma_\mathrm{j}^\mathrm{inner}(t_\mathrm{ev})$ during wet and dry conditions. On the other hand, the continuous soil moisture measurements cover a larger extent of the two studied catchments and different depths in the soil profile. Due to the higher variability at the catchment scale, the mean $SEM_\mathrm{cat.}^\mathrm{inter}$ is somewhat higher (1.5 vol%). The values obtained for $SEM_\mathrm{j}^\mathrm{inner}$ and $SEM_\mathrm{cat.}^\mathrm{inter}$ show that at an accepted uncertainty of the mean of ± 2 vol%, the number of 10 measurements per plot is sufficient. Furthermore, the

continuous measurements reveal the temporal evolution of soil moisture over the season and within events. The only drawback is the lack of continuous soil moisture estimates in the topsoil. The sampling at the plot scale and in nested catchments is considered to be a good approach to assess heterogeneity across scales and to cope with the change of scale problem (Braud et al., 2014).

This study's result of normally distributed soil moisture at the plot scale with a high spatial variability at scales smaller than

ten meters agrees well with other studies reviewed in Vereecken et al. (2014) and with results that Huza et al. (2014) obtained in the grasslands of the Gazel catchments. The latter authors also analyzed semi-variograms of six grasslands and report a very high nugget which indicates systematically high differences between measurements of less than 1 m distance. Unlike other studies that report positively or negatively skewed distributions for dry and wet conditions (Vereecken et al., 2014), the data of the present study is normally distributed during wet and dry conditions as well. The finding of a lower

variability during dry conditions (Huza et al., 2014; Vereecken et al., 2014) is not observed here ($R^2 = 0.04$), even though the range of plot mean soil moisture covers very diverse conditions. The differences of this study's results and the ones found by Huza et al. (2014) may be due to a different sampling locations as Huza et al. (2014) did not sample vineyards. If in this study's data set only grasslands are considered there is a slightly better but still poor correlation ($R^2 = 0.13$) between $\overline{\theta}_\mathrm{j}(t_\mathrm{ev})$ and $\sigma_\mathrm{j}^\mathrm{inner}(t_\mathrm{ev})$.

The observation that the probability density functions of soil moisture measurements on various plots in the catchment either agree with the normal distribution function or show a bi- or multi-modal behavior also agrees with other study's findings (Vereecken et al., 2014). In this study the inter-plot variability within one land use class $\sigma_\mathrm{lu}^\mathrm{inter}(t_\mathrm{ev})$ usually exceeds the inner-plot variability $\sigma_\mathrm{j}^\mathrm{inner}(t_\mathrm{ev})$. This is not consistent with findings of Huza et al. (2014) which may again be due to different sampling strategies: whereas Huza et al. (2014) conducted measurements along 50 m transects, for this study

random locations were sampled within one field in an area of ≈ 20 by 20 m. At scales larger than 10 m, they found a spatial structure revealed by a higher semi-variance at distances of more than 10 m in at least one of their transects.





The temporal dynamics with a quick response to rain events and the strong accordance between the evolution of soil moisture and discharge of small catchments is also observed by Braud et al. (2014) in the Valescure catchment. This catchment is also located in the Cévennes-Vivarais region and shows a similar dynamic of the interaction of soil moisture and discharge to the one observed here.

## 4.2 Comparison between land use classes and analysis of temporal stability

The results of this study indicate differences between grasslands and vineyards in the vertical soil profile, in the response of the profile to rain events, in percolation behavior and in the persistence of soil moisture conditions. These differences are most likely due to differences in soil texture, as vineyards are usually found on soils with higher clay content than the ones of the other land use types. The different percolation behavior of grasslands may also be due to differences in root structure as the dense grass roots may accelerate downward water movement along preferential flow paths more efficiently than the ones of the vine plants. However, there are no significant and systematic differences between the plot means of different land use classes. Thus, land use cannot be used as additional information to improve spatially distributed soil moisture estimation in the study site.

The cultivated field c1 shows a remarkable temporal stability of the difference of this plot's mean soil moisture and the catchment mean $\delta_{j,ev}$. This suggests that if the catchment mean has to be approximated by measurements in just one field, this one is the best choice (Vachaud et al., 1985; Vanderlinden et al., 2012). Other fields show, however, that $\delta_{j,ev}$ is not consistent in time. The observation that several sites change the sign of $\delta_{j,ev}$ between measurements was also made on the plot scale on a grassland, a field cultivated with wheat and an olive grove by Vachaud et al. (1985) and on the catchment scale on grasslands by Huza et al. (2014). Here, notably v1 and v2 are considerably wetter than the catchment mean throughout the autumn seasons of 2012 and 2013, dryer in 2014 and again wetter in 2015. Possible reasons include changes in cultivation. Especially tillage practices play an important role in the vineyards (not shown here). Therefore, conclusions based on this finding should be considered carefully. Moreover, the choice of the plot which best represents the catchment mean should include the temporal variability of $\delta_{j,ev}$ and should not be solely based on the minimal mean difference $\overline{\delta_j}$ which is in this case the one of v1 and v2.

## 4.3 Quantification of the hydrological response

Besides the extensive soil moisture data set used for this study, the available precipitation and discharge data at a high spatio-temporal resolution is a major asset that is necessary to understand the hydrological processes at small scales and during short time spans that lead to flash flood generation (Nord et al., 2017). It allows to calculate the event-based runoff coefficient $\phi_{ev}$ and to estimate its uncertainty. The main sources of its uncertainty comprise that of the stage discharge relation which is especially important for events with high discharge and which was assessed with the BaRatin framework, the uncertainty associated to the choice of the method used for hydrograph separation and the uncertainty of the catchment




mean precipitation. The latter source of uncertainty is not considered in this study. It stems from the rainfall measurements with tipping buckets and the interpolation between the rain gauges. Tipping buckets are known to underestimate precipitation at high intensities (Marsalek, 1981; Molini et al., 2005), thus, including radar data could improve the estimation of catchment mean rainfall even in relatively well gauged catchments such as the ones of the Gazel and Claduègne (Creutin
and Borga, 2003; Delrieu et al., 2014; Abon et al., 2015).

In this study, the uncertainty associated to the hydrograph separation method exceeds that of the stage-discharge relation. The high range and positive skewness of event-based runoff coefficients is consistent with other study's results (Merz et al., 2006; Blume et al., 2007; Merz and Blöschl, 2009; Norbiato et al., 2009; Marchi et al., 2010). The dependence of $\phi_{ev}$ on rain characteristics suggested by other authors (Merz et al., 2006; Norbiato et al., 2009) was not entirely confirmed in this study
as none of the rain characteristics examined here ($P_{cum}$, $I_\mu$, $I_{max,20}$) could explain more than 30 % of the variability in $\phi_{ev}$.

## 4.4 Validity of the hydrograph separation method

Each of the hydrograph separation methods used here has advantages and disadvantages. The method based on electric conductivity ($EC$) has a physically based foundation as it distinguishes components with different $EC$ and represents subsurface flow dynamics. This method could not be applied to both catchments because values for $EC_{ev}$ were only available
from Le Pradel in the south of the Gazel catchment and it is assumed that $EC_{ev}$ on the basaltic plateau differs considerably while this geology accounts for a large part of the Claduègne catchment. Furthermore, it is not possible to conduct a three component hydrograph decomposition with the available data, so unlike with the other methods, the fast reacting subsurface flow is considered to be baseflow. Thus, event discharge is underestimated.

Unlike the other methods, the Constant-k (CK) method offers a physical explanation for the end of event flow. The method
builds on the assumption of baseflow behaving like the slow responding outflow of a linear reservoir. For the discharge data of the Gazel, this method could not always be applied because of the low discharge that results in "steps" in the data and high noise so the threshold for defining that $k$ is constant as proposed by Blume et al. (2007) was never reached. An adjusted threshold yielded reasonable results for some but not all events.

The straight line method was rejected because it does not consider baseflow dynamics and the end of event flow has to be
determined arbitrarily. The filter methods have the advantage of being easy to apply to all data sets without further data treatment or demand of additional data but these methods are very sensitive to parameters such as the interval width (HySep filters) or the number of passes (RDF). The HySep filters were discarded because of the disagreement with the other methods. Thus, the RDF method was used for all further analyses because it correlates well with the EC and CK methods and can easily be applied to all events and both catchments. The number of passes had to be calibrated as suggested by
Ladson et al. (2013) in a manner that $\phi_{ev}$ is below 1 for all cases and that it is slightly higher than the value obtained with the EC method in order to compensate for the underestimation of event discharge. Nonetheless, underestimation of event discharge is still a source of uncertainty.



## 4.5 The impact of initial soil moisture on the hydrological response

The relation between $\phi_{ev}$ and $\tilde{\theta}_{ini}$ is not as clear as one might have expected from other studies results which suggest a dependence of $\phi_{ev}$ on $\tilde{\theta}_{ini}$. Moreover, both variables are still subject to large uncertainties. Catchment mean initial soil moisture $\tilde{\theta}_{ini}$ below a threshold of 34 vol% inhibits high $\phi_{ev}$. However, only three of the events considered here occur during such dry conditions, so further measurements would be useful to corroborate this finding. It is consistent with the observations made by Huza et al. (2014) in the Gazel catchment and Braud et al. (2014) in the Valescure catchment. The thresholds obtained by these authors (22 and 25 vol% respectively) are lower than the one obtained here. These differences might be due to different soil and land use features and different sampling designs. The values for $\tilde{\theta}_{ini}$ that Huza et al. (2014) used are obtained from satellite data, while this study uses in situ data from several land use classes. Moreover, a profile mean is considered here, while Huza et al. (2014) used only values of topsoil moisture. Furthermore, different methods were applied for hydrograph separation. Huza et al. (2014) used a method similar to the HySep 3 filter, which yielded different results than the other methods applied for this study.

The high range of $\phi_{ev}$ obtained at high $\tilde{\theta}_{ini}$ also agrees with findings of Huza et al. (2014). It indicates, that the hydrological response is influenced by other factors as well. The parameters describing the impact of meteorological forcing ($P_{cum}$, $I_\mu$ and $I_{max,20}$) neither explain that variability. When only events with high cumulative precipitation are considered, the range is still very high. Results obtained in virtual experiments by Merz and Plate (1997) and Zehe and Blöschl (2004) showed that spatial patterns of soil moisture and threshold effects strongly impact the runoff response. The latter authors show, that especially during initial moisture conditions close to the threshold, the runoff response depends strongly on the resampling of spatially distributed soil moisture. Therefore, actual, small-scale soil moisture patterns that control connectivity of pathways but are not reflected in the catchment mean value used for Fig. 11 are a possible explanation for the very diverse runoff behavior. Additionally, subsurface flow along preferential flowpaths can contribute to high $\phi_{ev}$ and Hortonian overland flow is not directly related to $\tilde{\theta}_{ini}$ but produces a substantial proportion of event flow.

The analysis of the baseflow dynamics as obtained by the hydrograph separation with $EC$ shows that for almost all events there is a quick response of the subsurface stormflow which constitutes a large proportion of total event flow. This indicates that subsurface stormflow plays an important role in the Gazel catchment and should be considered thoroughly. The maximum $EC$ signal during discharge events $\Delta EC_{max}$ varies considerably between events. Figure 13 shows that possible reasons include high $\tilde{\theta}_{ini}$ and high $P_{cum}$ which might be indicators for saturation excess overland flow but also high $I_{max,20}$, a possible indicator of infiltration excess Hortonian overland flow. Even though $I_{max,20}$ is usually below 10 mm h$^{-1}$ the rather low values for saturated conductivity $K$s measured by Braud and Vandervaere (2015) indicate that rain intensity can locally exceed infiltration capacities.



## 5 Conclusions

This study aimed at assessing the influence of initial soil moisture on the hydrological response in a flash-flood prone area in southern France. To this end, two research questions were addressed and exemplarily examined in the nested Gazel (3.4 km$^2$) and Claduègne (43 km$^2$) catchments: (1) How can soil moisture heterogeneity in time and space be described and how does soil moisture correlate with land use? (2) What is the relationship between initial soil moisture $\tilde{\theta}_{ini}$ and the hydrological response quantified as the event-based runoff coefficient $\phi_{ev}$?

The main findings of this study related to the first research questions are:

(1.1) Spatial variability of soil moisture at the plot scale is very high and the probability density function (pdf) of plot-scale soil moisture measurements usually follows a normal distribution. Soil moisture at the catchment scale is also very variable, and the pdf of the measurements often resembles a bi- or multi-modal distribution, which is an indicator for inter-plot variability. This is corroborated as inter-plot standard deviation $\sigma^{inter}$ exceeds inner-plot standard deviation $\sigma^{inner}$.

(1.2) There are differences between land use classes in the vertical soil moisture profile, notably that the grasslands have a relatively homogeneous initial soil moisture profile whereas the vineyards have a curved profile. There are further differences in wetting behavior and in percolation, but no significant and systematic differences in catchment mean soil moisture values between land use classes. Hence, land use cannot be used as an auxiliary variable to determine the spatial distribution of soil moisture. Between land use standard deviation $\sigma_{lu}^{betw}$ exceeds neither $\sigma^{inter}$ nor $\sigma^{inner}$.

(1.3) There is one plot, c1, with remarkable temporal stability of the spatial difference between plot mean and catchment mean. Thus, this field should be opted for, if the catchment mean had to be assessed from measurements in just one plot. However, none of the other plots shows this temporal stability.

(1.4) The temporal dynamics of soil moisture show a seasonal trend, a quick reaction to rain events and fast percolation to lower layers. During dry periods moisture conditions can persist for a long time with just a slow decrease in soil moisture.

The sampling design applied for this study allowed a detailed characterization of soil moisture heterogeneity across scales as well as the assessment of temporal dynamics. The catchment mean soil moisture was derived with a mean standard error of the catchment mean of 1.3 vol% or 1.5 vol% for the Gazel and Claduègne catchments respectively.

Main findings concerning the impact of initial soil moisture on the hydrological response quantified with the event-based runoff coefficient $\phi_{ev}$ are:

(2.1) $\phi_{ev}$ obtained with different hydrograph separation methods can differ considerably, but results obtained with EC, CK, SL and RDF methods correlate well. The RDF method was preferred for this study because it is easy to apply and because of the good correlation with the more physically based methods EC and CK which could not be applied to all events and both catchments.

(2.2) There is a week correlation between $\phi_{ev}$ and cumulative event precipitation $P_{cum}$, mean rain intensity $I_\mu$ and maximum 20 min rain intensity $I_{max,20}$ ($R^2 = 0.24$, 0.17 or 0.27 respectively).





(2.3) The hydrological response depends on initial soil moisture $\tilde{\theta}_{\mathrm{ini}}$: below a threshold of 34 vol%, $\phi_{\mathrm{ev}}$ remains very low, even during high precipitation events. However, there is a large scatter in $\phi_{\mathrm{ev}}$ above that threshold indicating that other factors and processes also have an important impact on $\phi_{\mathrm{ev}}$. The threshold is identical for both catchments which indicates that at this study's site it might be scale invariant.

(2.4) Regarding the seasonal and within event evolution of soil moisture and discharge shows that discharge peaks of two considered events did not follow the peaks in rainfall, but a second, smaller rain impulse, while the rainfall peaks lead to a considerable refilling of soil water storage.

(2.5) Some events reacted with a strong signal in electrical conductivity, indicating an important contribution of overland flow. There are several possible explanations for high $EC$ signals, such as high $P_{\mathrm{cum}}$, $\tilde{\theta}_{\mathrm{ini}}$ or $I_{\mathrm{max,20}}$. Many events are

10 dominated by a considerable contribution of subsurface flow to total event flow and show a fast response in subsurface stormflow to rain events.

These results indicate, that $\tilde{\theta}_{\mathrm{ini}}$ does impact the hydrological response. For single events $\phi_{\mathrm{ev}}$ or $EC$ signals can be attributed to $\tilde{\theta}_{\mathrm{ini}}$, $P_{\mathrm{cum}}$ or $I_{\mathrm{max,20}}$. However, these results cannot be generalized and no systematic and unequivocal relationship between $\tilde{\theta}_{\mathrm{ini}}$ and $\phi_{\mathrm{ev}}$ was found. Thus, the second research question could only partly be answered. Even though the present data set is

15 exceptionally detailed, there still is substantial uncertainty in the values for $\tilde{\theta}_{\mathrm{ini}}$, $P_{\mathrm{cum}}$ and cumulative event flow $Q_{\mathrm{ev,cum}}$.

The results of this study partly confirm the suggestions of other authors to consider estimates of initial soil moisture in flash flood warning based on the dependence of $\phi_{\mathrm{ev}}$ on $\tilde{\theta}_{\mathrm{ini}}$. Threshold-based warning systems are advocated e.g. by Norbiato et al. (2008). Including a threshold value for initial soil moisture could prevent false positive flash flood warnings in cases when high precipitation is expected under dry initial catchment conditions, while above-threshold soil moisture in combination

with high precipitation increases the likelihood of high runoff coefficients. This threshold seems not to be scale-dependent. However, the threshold values differ between catchments and depend to a high degree on the methodology to determine it as indicated by the different values in this study and the one by Huza et al. (2014). Furthermore, there are high data requirements to determine such thresholds and it is not known whether they can be transferred from one catchment to another, so it is not applicable for operational flash-flood warning. Moreover, the high scatter of $\phi_{\mathrm{ev}}$ under high initial soil

moisture conditions suggests that the relation between $\phi_{\mathrm{ev}}$ and $\tilde{\theta}_{\mathrm{ini}}$ is very complex and depends on other factors and processes that are still insufficiently understood. Thus, the impact of soil moisture on the hydrological response during wet catchment conditions cannot be generalized based on the results obtained here.

In the Gazel catchment subsurface flow seems to constitute an important part of event flow, so further information on preferential flow path that could be obtained from tracer experiments would be helpful to understand the hydrological

behavior of the catchment. Further research could also focus on elaborating regression analysis methods that systematically examine different controls on $\phi_{\mathrm{ev}}$ such as meteorological forcing as well as $\tilde{\theta}_{\mathrm{ini}}$ and their interactions.





## Author contributions

Jean-Pierre Vandervaere and Isabella Zin conceptualized the study and Magdalena Uber carried out data analysis and its interpretation. Jean-Pierre Vandervaere, Isabelle Braud, Cédric Legoût, Gilles Molinié and Guillaume Nord were principal investigators in the FloodScale project responsible for instrumentation and criticized and provided the data used for this study. All authors contributed to data analysis and interpretation of results. Magdalena Uber prepared the manuscript with contributions from all co-authors. The authors declare that they have no conflict of interest.

## Acknowledgements

This work was funded by the French National Research Agency (ANR) via the FloodScale project under contract no. ANR 2011 BS56 027 which contributes to the HyMeX program. OHM-CV is supported by the Institut National des Sciences de l'Univers (INSU/CNRS), the French Ministry for Education and Research, the Environment Research Cluster of the Rhône-Alpes Region, the Observatoire des Sciences de l'Univers de Grenoble (OSUG) and the SOERE Réseau des Bassins Versants (Alliance Allenvi). Magdalena Uber received financial support from German Academic Exchange Service (DAAD). The authors want to thank Brice Boudevillain, Sahar Hachani, Simon Gérard and Cindy Nicoud for their help during field work.

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





**Figures and Tables**

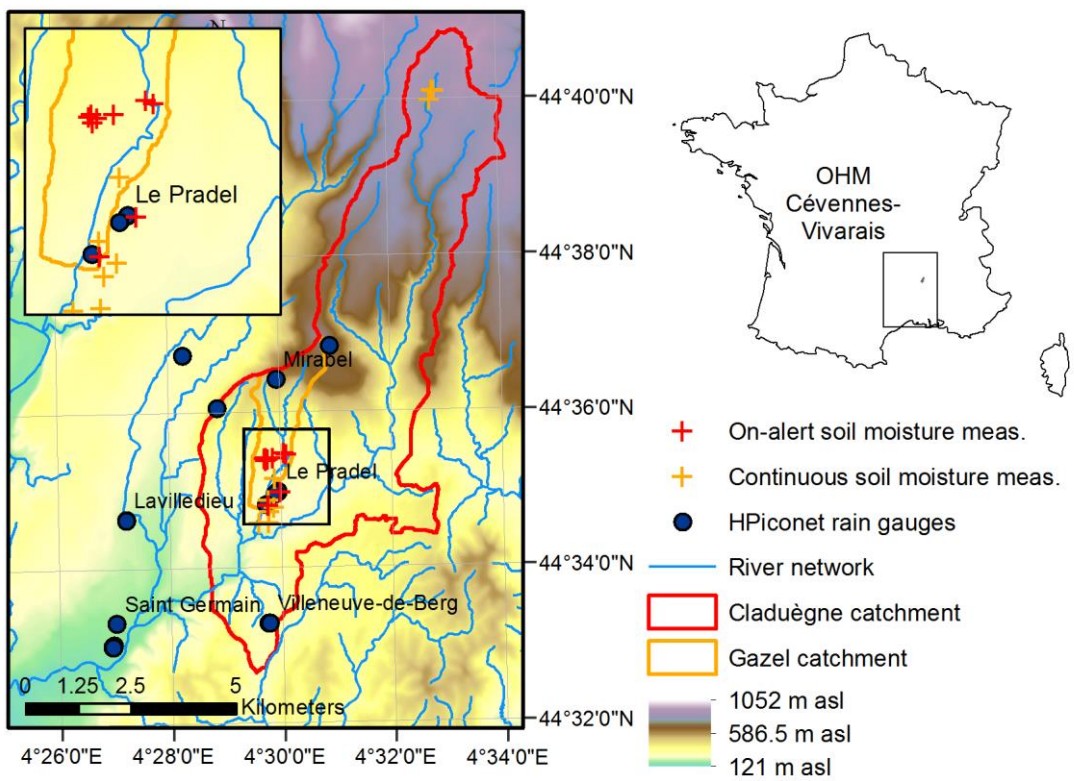

Figure 1: Location of the study site and measurement network.

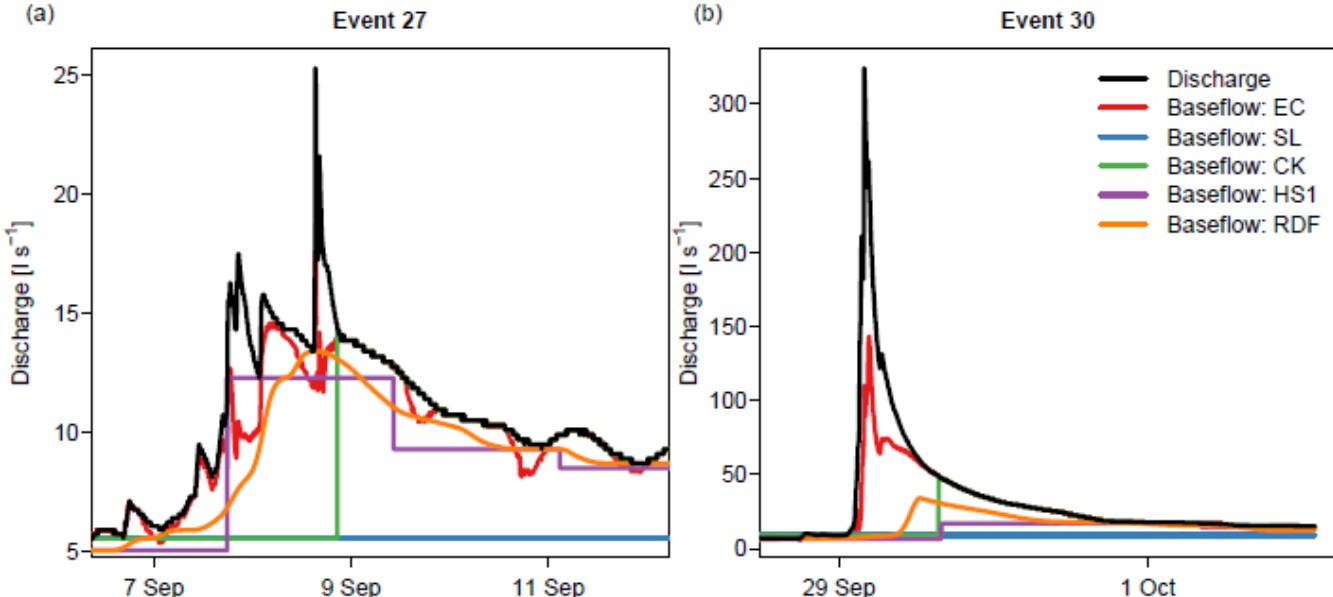

Figure 2: Hydrograph separation into baseflow and event flow of two different events with different methods: electric conductivity (EC), straight line (SL), constant-K (CK), HySep filter 1 (HS1) and recursive digital filter (RDF). See the text for descriptions and references.





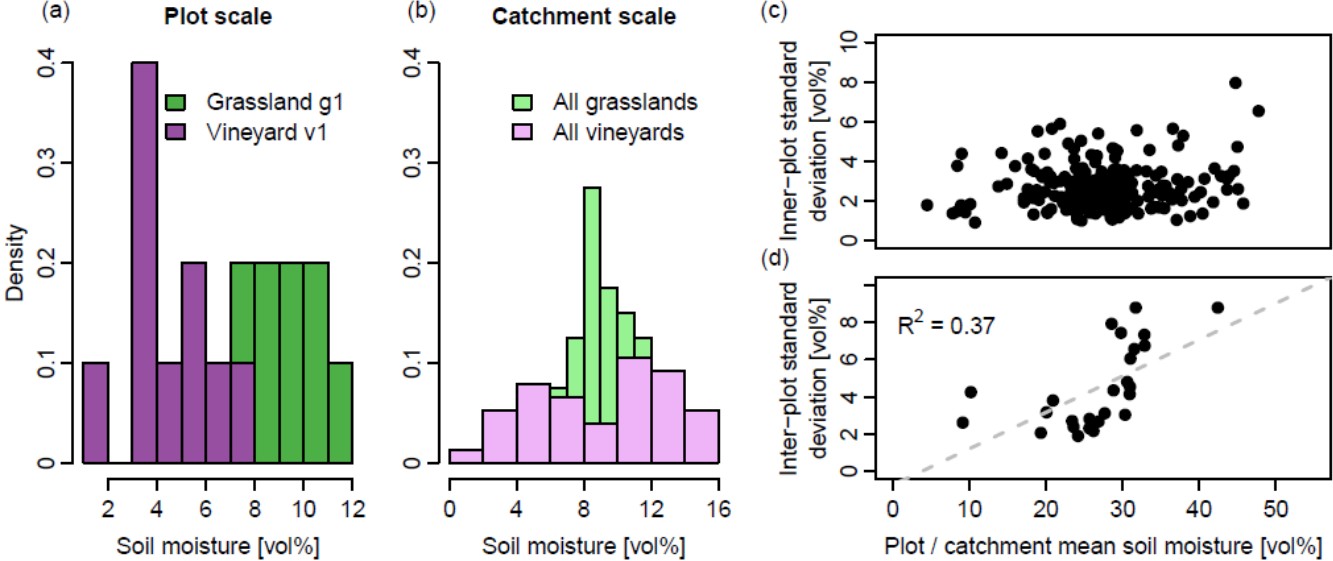

Figure 3: (a) Normalized histogram of soil moisture measurements within one grassland and one vineyard plot and (b) at the catchment scale within the land use classes grassland and vineyard during the first on-alert measurement in 2012. Figure (c) shows the relationship between plot mean soil moisture $\overline{\theta}_j(t_{ev})$ and inner-plot standard deviation $\sigma_j^{inner}(t_{ev})$ as calculated

5    with Eq. 1 and Eq. 6. Figure (d) shows the same at the catchment scale for catchment mean soil moisture $\overline{\overline{\theta}}_{ev}$ and inter-plot standard deviation $\sigma_{cat.}^{inter}(t_{ev})$ as calculated with Eq. 3 and Eq. 8.




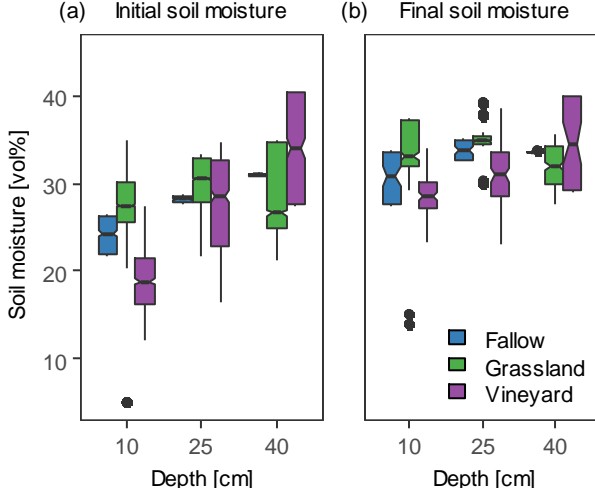

Figure 4: Initial (a) and final (b) soil moisture profile in plots of different land use during event 27 (06–09 September 2013).





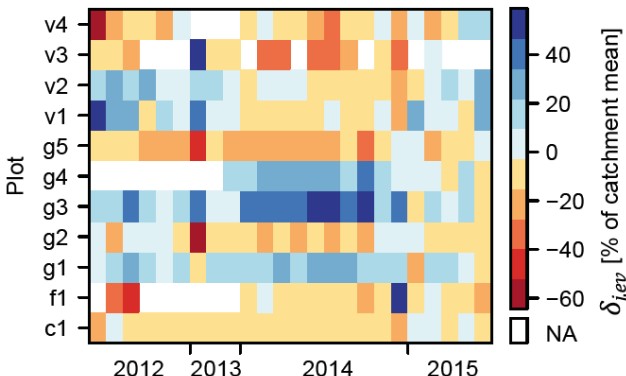

Figure 5: Temporal stability (Vachaud et al., 1985): the relative difference between the plot mean and the catchment mean $\delta_{j,ev}$ for each on-alert measurement and each plot. Note that the time axis represents a sequence of events, no equidistant time line. Blue squares show plots with plot means that exceed the catchment mean, red squares those with plot means below the catchment mean, white squares indicate plots that were not sampled during the respective measurement.





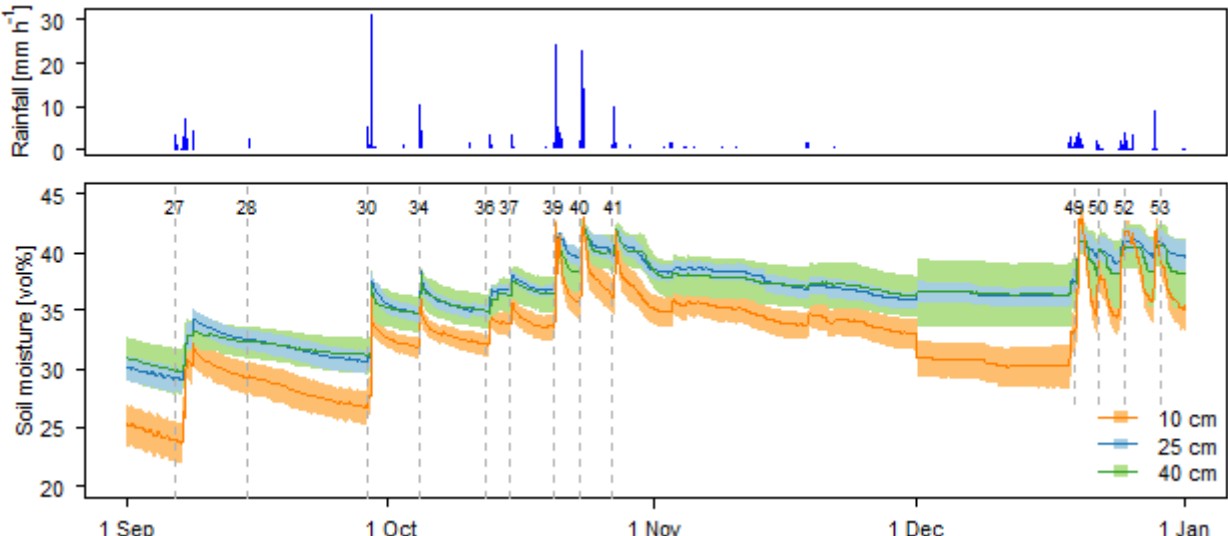

Figure 6: Rainfall and soil moisture in autumn 2013 measured in 10, 25 and 40 cm depth. The line represents the catchment mean in the respective depth and the shaded area the mean $\pm SEM_{cat.}^{inter}$. The dashed lines represent the onsets of several rain events and the labels refer to the event numbers as in Table 3 and Table 4. Note that on 27 November 2013 one of the probes

5   stopped working which is the reason for the step in the data and the higher $SEM_{cat.}^{inter}$.




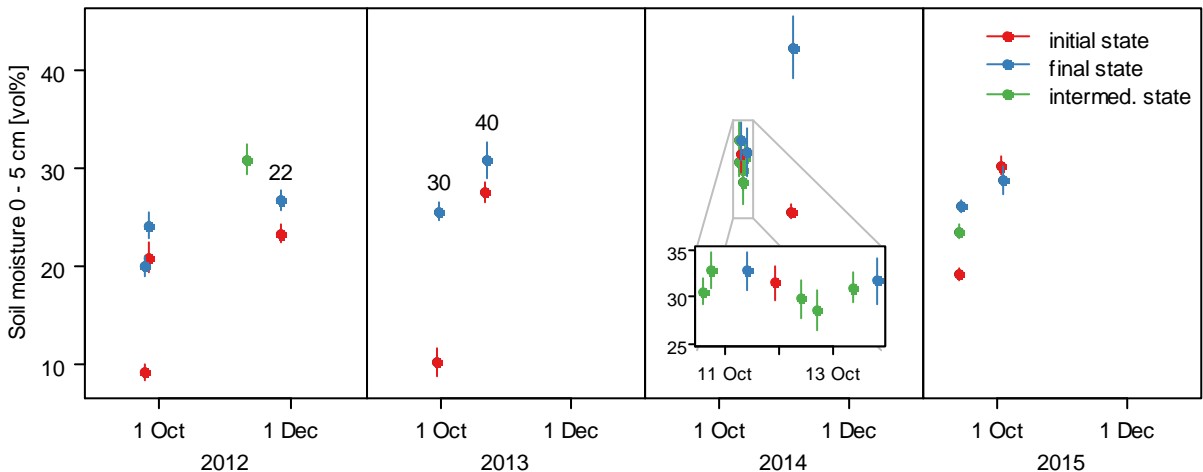

Figure 7: Soil moisture in the topsoil (0–5 cm) measured on alert basis before and after major rain events in autumn seasons in 2012–2015. The points show catchment mean values, the lines the range of the mean $\pm SEM_{cat.}^{inter}$. The numbers above selected events give the event number as in Table 3 and Table 4 and as referred to in the text.





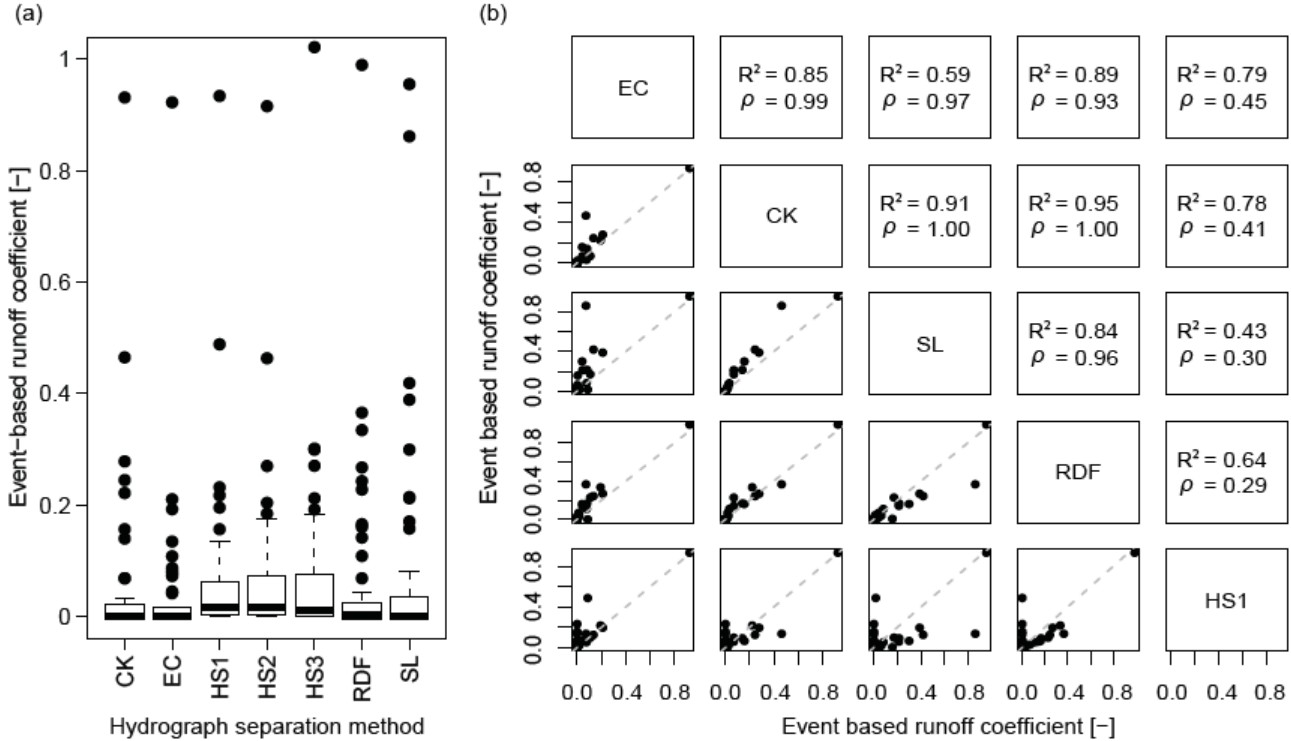

Figure 8: (a) Differences of event-based runoff coefficients $\phi_{ev}$ calculated for 54 rain events in the Gazel catchment in autumn 2012 and autumn 2013 derived with different methods for hydrograph separation. (b) Correlation between different methods. The upper panel gives the coefficient of determination ($R^2$) and Spearman's rank correlation coefficient $\rho$, the dashed line in the lower panel is the line of identity. The methods used for hydrograph separation are described in Sec. 2.5.2: constant-k (CK), electric conductivity (EC), Hysep filter with fixed or sliding interval (HS1–HS2), Hysep filter with local minima algorithm (HS3), recursive digital filter (RDF) and straight line (SL).

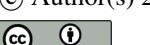



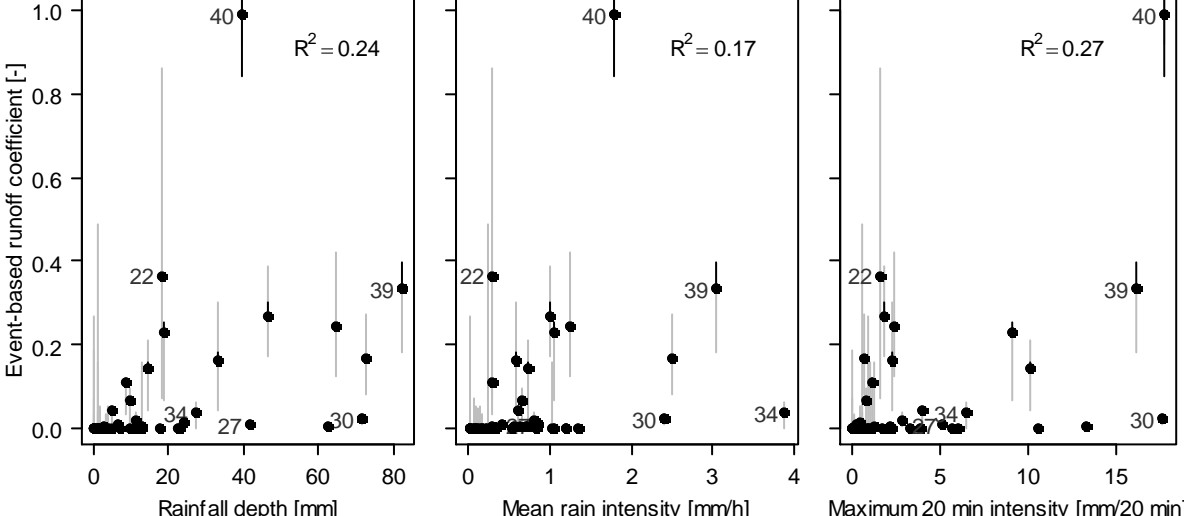

Figure 9: Correlation of three variables describing meteorological forcing with event-based runoff coefficients $\phi_{ev}$ of the Gazel catchment calculated with the recursive digital filter method. The lines represent the uncertainty associated to the hydrograph separation method (gray vertical lines: range of $\phi_{ev}$ calculated with the seven different methods) and the stage-discharge relation (black vertical lines: range between $\phi_{ev}$ calculated with the 5 % and the 95 % confidence interval of discharge obtained

with the BaRatin framework). The point labels give the numbers of selected events as in Table 3 and Table 4 and described in the text.





Figure 10: Hyetographs, hydrographs of total discharge and evolution of soil moisture in the Gazel catchment during four different events in 2013. The event-based runoff coefficient $\phi_{ev}$ is also given for all events. The representation of soil moisture gives the mean $\pm SEM_{\text{cat.}}^{\text{inter}}$ in the respective depth and the one of discharge the best estimate $\pm$ the uncertainty of the stage-discharge relation.





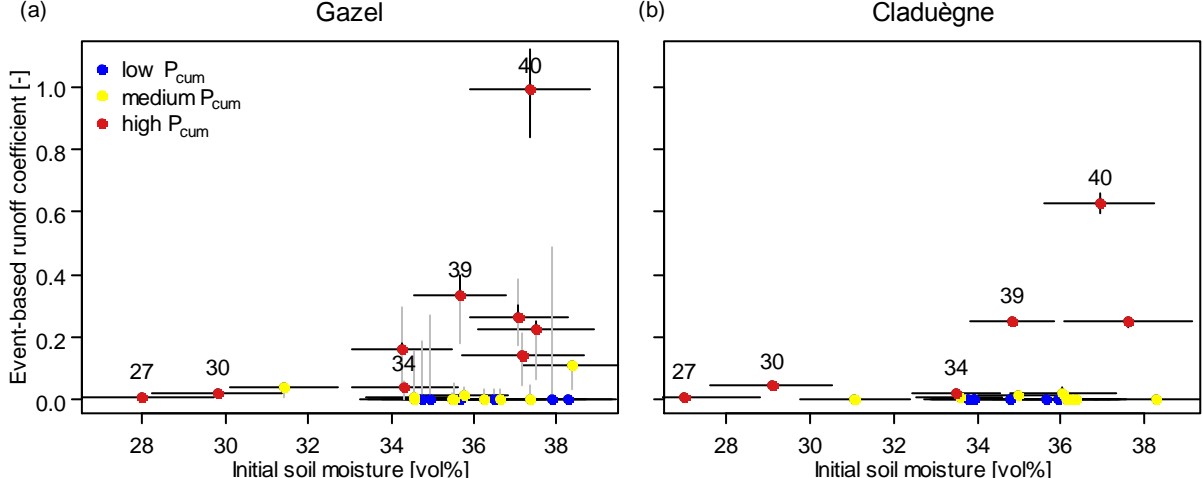

Figure 11: Relationship between initial soil moisture $\tilde{\theta}_{ini}$ and event-based runoff coefficients $\phi_{ev}$ in the Gazel (a) and Claduègne (b) catchments. On the x-axis the point represents the profile mean initial soil moisture and the horizontal line the range of the mean $\pm SEM_{cat.}^{inter}$. On the y-axis the point represents $\phi_{ev}$ calculated with the recursive digital filter method and the line the uncertainty as in Fig. 9. The color of the points indicates whether cumulative precipitation is low ($P_{cum} < 1.5$ mm), medium (1.5 mm $< P_{cum} < 13$ mm) or high ($P_{cum} > 13$ mm).




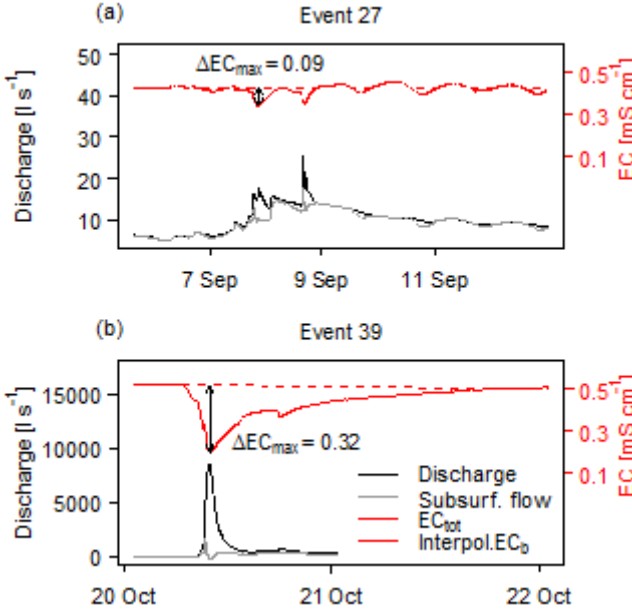

Figure 12: Streamflow signal of electric conductivity (*EC*) during two events in the Gazel catchment and hydrograph separation into surface and subsurface flow with the electric conductivity method. The maximum signal in streamflow *EC*, $\Delta EC_{max}$ that is calculated as the maximum of the difference between streamflow *EC*, $EC_{tot}$ and interpolated baseflow *EC*, $EC_b$ differs substantially between events.



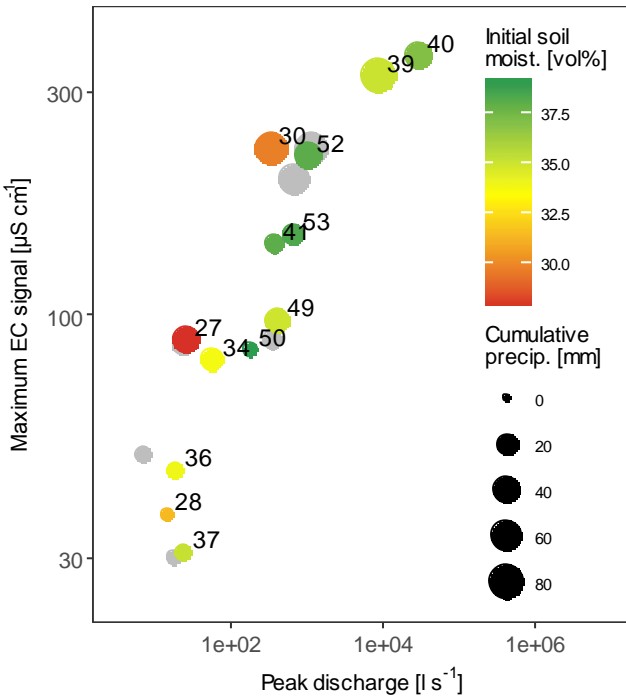

Figure 13: Log-log scatter plot of the maximum electric conductivity signal $\Delta EC_{max}$ of rain events in the Gazel catchment in 2013 and 2012 (gray dots: no data for $\tilde{\theta}_{ini}$ available) versus peak discharge $Q_p$, initial soil moisture $\tilde{\theta}_{ini}$, and cumulative Precipitation $P_{cum}$.





Table 1: Spatial variability of soil moisture at the plot scale (mean of all events calculated for all plots: mean $\sigma_j^{inner}(t_{ev})$, for the grassland plots: mean $\sigma_{j\in g}^{inner}(t_{ev})$ and the vineyard plots: mean $\sigma_{j\in v}^{inner}(t_{ev})$ calculated with Eq. 6) and at the catchment scale (mean inter plot variability of the grassland plots: mean $\sigma_g^{inter}(t_{ev})$ and the vineyard plots mean $\sigma_v^{inter}(t_{ev})$ calculated with Eq. 7 as well as between land use class variability mean $\sigma^{betw.}(t_{ev})$ calculated with Eq. 8) determined at different depth with the two measuring schemes.

|  | Initial states |  |  |  | Final states |  |  |  |
|---|---|---|---|---|---|---|---|---|
|  | 0-5 cm | 10 cm | 25 cm | 40 cm | 0-5 cm | 10 cm | 25 cm | 40 cm |
| Mean $\sigma_j^{inner}(t_{ev})$ | 2.62 | 2.77 | 1.85 | NA | 2.91 | 2.71 | 1.88 | NA |
| Mean $\sigma_{j\in g}^{inner}(t_{ev})$ | 2.77 | 2.18 | 1.77 | NA | 2.80 | 2.18 | 1.78 | NA |
| Mean $\sigma_{j\in v}^{inner}(t_{ev})$ | 2.49 | 3.15 | 2.24 | NA | 2.84 | 3.15 | 2.24 | NA |
| Mean $\sigma_g^{inter}(t_{ev})$ | 3.48 | 4.49 | 2.07 | 4.40 | 3.70 | 4.26 | 2.04 | 4.30 |
| Mean $\sigma_v^{inter}(t_{ev})$ | 2.63 | 2.26 | 5.36 | 6.81 | 2.11 | 2.39 | 5.31 | 6.65 |
| Mean $\sigma^{betw.}(t_{ev})$ | 2.12 | 2.20 | 1.14 | 1.71 | 3.78 | 1.98 | 1.10 | 1.55 |





Table 2: Differences between land use classes: results of Student t-tests under significance level α = 0.05. The upper panel gives the ratio of the number of rejections of the null hypothesis (true difference in means is equal to zero) to the total number of tests conducted. The lower panel states whether the found differences are systematic (T) or not (F)

|  | Grassland | Vineyard | Cultivated | Fallow |
|---|---|---|---|---|
| Grassland | - | 15 / 26 | 17 / 24 | 12 / 17 |
| Vineyard | F | - | 11 / 24 | 8 / 17 |
| Cultivated | T | F | - | 4 / 17 |
| Fallow | F | F | F | - |





Table 3: Rainfall, soil moisture and discharge characteristics of selected rain events in autumn 2013 in the Gazel catchment: beginning of the rain event, cumulative precipitation ($P_{cum}$), maximum 20-min rain intensity ($I_{max,20}$), mean intensity ($I_\mu$), initial soi moisture ($\tilde{\theta}_{ini}$), final soil moisture ($\tilde{\theta}_{fin}$), standard error of the catchment mean ($SEM_{cat.}^{inter}$) during initial and final stage, soil storage change ($\Delta S$), peak discharge ($Q_p$), cumulative total discharge ($Q_{tot,cum}$), cumulative event discharge ($Q_{ev,cum}$), event-based runoff coefficient calculated with the recursive digital filter method ($\phi_{ev}$) and maximum signal in electric conductivity ($\Delta EC_{max}$).

|  | Rainfall |  |  |  | Soil Moisture |  |  |  |  | Discharge |  |  |  |
|---|---|---|---|---|---|---|---|---|---|---|---|---|---|
| Ev. # | Beg. Rain DD-MM hh:mm | $P_{cum}$ mm | $I_{max,20}$ mm h$^{-1}$ | $I_\mu$ mm h$^{-1}$ | $\tilde{\theta}_{ini}$ vol% | $SEM_{cat.}^{inter}$ vol% | $\tilde{\theta}_{fin}$ vol% | $SEM_{cat.}^{inter}$ vol% | $\Delta S$ mm | $Q_p$ l s$^{-1}$ | $Q_{ev,cum}$ mm | $\phi_{ev}$ - | $\Delta EC_{max}$ µS cm$^{-1}$ |
| 27 | 06-09 16:06 | 41.59 | 33.30 | 0.85 | 27.99 | 1.73 | 32.69 | 1.07 | 28.17 | 25 | 0.34 | 0.01 | 88 |
| 28 | 15-09 00:23 | 4.82 | 24.00 | 0.60 | 31.40 | 1.31 | 31.34 | 1.38 | 0.00 | 14 | 0.20 | 0.04 | 37 |
| 30 | 28-09 17:09 | 71.73 | 59.70 | 2.39 | 29.80 | 1.57 | 35.20 | 1.45 | 32.43 | 324 | 1.73 | 0.02 | 228 |
| 34 | 04-10 15:39 | 27.20 | 25.50 | 3.89 | 34.33 | 1.29 | 36.43 | 1.25 | 12.62 | 58 | 1.03 | 0.04 | 80 |
| 36 | 12-10 05:12 | 12.86 | 4.50 | 0.71 | 34.55 | 1.15 | 35.91 | 1.02 | 8.12 | 18 | 0.07 | 0.01 | 46 |
| 37 | 15-10 03:46 | 11.29 | 9.00 | 0.81 | 35.76 | 1.07 | 36.54 | 1.07 | 4.72 | 23 | 0.21 | 0.02 | 31 |
| 39 | 20-10 02:41 | 82.20 | 53.40 | 3.04 | 35.65 | 1.10 | 38.12 | 1.41 | 14.83 | 8660 | 27.49 | 0.33 | 324 |
| 40 | 23-10 01:01 | 39.31 | 60.00 | 1.79 | 37.34 | 1.44 | 38.40 | 1.54 | 6.33 | 30096 | 38.91 | 0.99 | 357 |
| 41 | 27-10 03:37 | 14.64 | 37.20 | 0.73 | 37.17 | 1.46 | 38.09 | 1.52 | 5.54 | 361 | 2.07 | 0.14 | 142 |
| 49 | 18-12 06:52 | 33.09 | 8.70 | 0.57 | 34.26 | 1.20 | 37.90 | 1.32 | 21.82 | 402 | 5.30 | 0.16 | 97 |
| 50 | 21-12 03:11 | 8.73 | 4.20 | 0.27 | 38.39 | 1.23 | 37.78 | 1.12 | 0.00 | 175 | 0.95 | 0.11 | 84 |
| 52 | 24-12 01:15 | 46.33 | 7.50 | 0.99 | 37.09 | 1.17 | 39.83 | 1.08 | 16.46 | 1047 | 12.38 | 0.27 | 218 |
| 53 | 28-12 04:51 | 18.70 | 32.70 | 1.04 | 37.52 | 1.40 | 39.25 | 1.04 | 10.41 | 654 | 4.25 | 0.23 | 149 |





Table 4: Rainfall, soil moisture and discharge characteristics of selected rain events in autumn 2013 in the Claduègne catchment, abbreviations as in Table 3.

| Ev. # | Rainfall Beg. Rain DD-MM hh:mm | $P_{cum}$ mm | $I_{max,20}$ mm h$^{-1}$ | $I_\mu$ mm h$^{-1}$ | Soil Moisture $\tilde{\theta}_{ini}$ vol% | $SEM_{cat.}^{inter}$ vol% | $\tilde{\theta}_{fin}$ vol% | $SEM_{cat.}^{inter}$ vol% | $\Delta S$ mm | Discharge $Q_p$ m$^3$ s$^{-1}$ | $Q_{ev,cum}$ mm | $\phi_{ev}$ - | $\Delta EC_{max}$ µS cm$^{-1}$ |
|---|---|---|---|---|---|---|---|---|---|---|---|---|---|
| 27 | 06-09 15:41 | 43.26 | 33.3 | 0.88 | 27.00 | 1.38 | 32.62 | 1.16 | 29.05 | 0.19 | 0.38 | 0.01 | 44 |
| 28 | 15-09 00:23 | 3.61 | 26.7 | 0.45 | 31.09 | 1.32 | 31.00 | 1.32 | 0.00 | NA | NA | NA | 11 |
| 30 | 28-09 17:04 | 77.71 | 63 | 2.59 | 29.09 | 1.49 | 34.49 | 1.24 | 29.93 | 16.39 | 3.16 | 0.04 | 273 |
| 34 | 04-10 15:33 | 28.09 | 27 | 4.01 | 33.48 | 1.53 | 35.46 | 1.44 | 11.39 | 0.59 | 0.66 | 0.02 | 54 |
| 36 | 12-10 05:12 | 12.81 | 4.5 | 0.71 | 33.58 | 1.53 | 35.20 | 1.46 | 8.42 | 0.13 | 0.10 | 0.01 | 19 |
| 37 | 15-10 03:46 | 11.51 | 9 | 0.82 | 34.98 | 1.55 | 35.67 | 1.52 | 4.34 | 0.16 | 0.19 | 0.02 | 26 |
| 39 | 20-10 02:41 | 83.67 | 53.4 | 3.10 | 34.83 | 1.63 | 37.54 | 1.51 | 15.33 | 54.64 | 20.82 | 0.25 | 177 |
| 40 | 23-10 01:01 | 51.01 | 60 | 2.32 | 36.92 | 1.69 | 38.75 | 1.60 | 11.37 | 60.76 | 36.37 | 0.93 | 115 |
| 41 | 27-10 03:37 | 22.06 | 52.5 | 1.10 | 37.62 | 1.67 | 38.47 | 1.64 | 4.40 | 12.08 | 5.44 | 0.37 | 22 |
| 49 | 18-12 06:52 | 67.91 | 8.7 | 1.17 | 34.40 | 1.61 | 38.22 | 1.44 | 21.59 | NA | NA | NA | NA |
| 50 | 21-12 03:11 | 10.98 | 4.2 | 0.34 | 38.70 | 1.61 | 38.40 | 1.63 | 0.00 | NA | NA | NA | NA |
| 52 | 24-12 01:15 | 51.24 | 7.5 | 1.09 | 37.70 | 1.68 | 40.29 | 1.58 | 14.30 | NA | NA | NA | NA |
| 53 | 28-12 04:51 | 19.65 | 32.7 | 1.09 | 38.09 | 1.66 | 39.68 | 1.60 | 9.56 | NA | NA | NA | NA |