# Peer review of "How does initial soil moisture influence the hydrological response? A case study from southern France"

_Hydrology and Earth System Sciences, 2018_

## Short Comment (SC1) · 13 Mar 2018

**Short Comments**

I quickly read the paper by Uber et al. as I have been working for several years on the role of initial soil moisture conditions on runoff generation. I appreciate the extensive analysis the authors have carried out in their study with a good experimental setup. However, I found strange that the authors didn't mention many papers on the same topic published in the scientific literature. Of course, I have authored (and co-authored) some of these studies, but I want to underline that I do not want the authors cite our

papers. I simply would like that the results of this study were considered in the context of current knowledge we have gained on the role of soil moisture for runoff generation. Some of previous studies have carried out a very similar analysis, i.e., by comparing ground (and satellite) soil moisture observations with the hydrologic response at basin scale for different flood events. I added below a (not exhaustive) list of references I'd like the authors consider.

**References (not exhaustive)**

Beck, H. E., R. A. M de Jeu, J. Schellekens , A. I. J. M van Dijk, L. A. Bruijnzeel, (2009), Improving Curve Number Based Storm Runoff Estimates Using Soil Moisture Proxies, Selected Topics in Applied Earth Observations and Remote Sensing, IEEE Journal of selected topics in applied earth observations and remote sensing, (2), 250-259.

Berthet, L., Andréassian, V., Perrin, C. and Javelle, P. (2009). How crucial is it to account for the Antecedent Moisture Conditions in flood forecasting? Comparison of event-based and continuous approaches on 178 catchments. Hydrol. Earth Syst. Sci., 13, 819-831.

Brocca, L., Melone, F., Moramarco, T., Morbidelli, R. (2009). Antecedent wetness conditions based on ERS scatterometer data. Journal of Hydrology, 364 (1-2), 73-87.

Brocca, L., Melone, F., Moramarco, T., Singh, V.P. (2009). Assimilation of observed soil moisture data in storm rainfall-runoff modelling. Journal of Hydrologic Engineering, 14 (2), 153-165.

Brocca, L., Melone, F., Moramarco, T., Penna, D., Borga, M., Matgen, P., Gumuzzio, A., Martinez-Fernández, J., Wagner, W. (2013). Detecting threshold hydrological response through satellite soil moisture data. Die Bodenkultur, 64(3-4), 7-12.

Crow, W.T., Bindlish, R. and Jackson, T.J. (2005) The added value of spaceborne passive microwave soil moisture retrievals for forecasting rainfall-runoff ratio partitioning.

Geophys. Res. Lett., 32, L18401.

Crow, W. T., Chen, F., Reichle, R. H., Liu, Q. (2017). L‐band microwave remote sensing and land data assimilation improve the representation of pre‐storm soil moisture conditions for hydrologic forecasting. Geophysical Research Letters, doi: 10.1002/2017GL073642.

Huang, M., Gallichand, J., Dong, C., Wang, Z. and Shao, M. (2007). Use of soil moisture data and curve number method for estimating runoff in the Loess Plateau of China. Hydrol. Process., 21(11), 1471-1481.

Massari, C., Brocca, L., Barbetta, S., Papathanasiou, C., Mimikou, M., Moramarco, T. (2014). Using globally available soil moisture indicators for flood modelling in Mediterranean catchments. Hydrology and Earth System Sciences, 18, 839-853, doi:10.5194/hess-18-839-2014.

Massari, C., Brocca, L., Moramarco, T., Tramblay, Y., Didon Lescot, J.-F. (2014). Potential of soil moisture observations in flood modelling: estimating initial conditions and correcting rainfall. Advances in Water Resources, 74, 44-53, doi:10.1016/j.advwatres.2014.08.004.

Massari, C., Brocca, L., Ciabatta, L., Moramarco, T., Gabellani, S., Albergel, C., de Rosnay, P., Puca, S., Wagner, W. (2015). The use of H-SAF soil moisture products for operational hydrology: flood modelling over Italy. Hydrology, 2(1), 2-22, doi:10.3390/hydrology2010002.

Morbidelli, R., Corradini, C., Saltalippi, C., Flammini, A., Brocca, L., Govindaraju, R.S. (2016). An investigation of the effects of spatial heterogeneity of initial soil moisture content on surface runoff simulation at a small watershed scale. Journal of Hydrology, 539, 589-598.

Penna, D., Tromp-van Meerveld, H. J., Gobbi, A., Borga, M., Dalla Fontana, G. (2011). The influence of soil moisture on threshold runoff generation processes in an alpine

headwater catchment. Hydrol. Earth Syst. Sci, 15, 689-702.

Tramblay, Y., Bouvier, C., Martin, C., Didon-Lescot, J. F., Todorovik, D., Domergue, J. M. (2010). Assessment of initial soil moisture conditions for event-based rainfall–runoff modelling. Journal of Hydrology, 387(3-4), 176-187.

Tramblay, Y., Bouaicha, R., Brocca, L., Dorigo, W., Bouvier, C., Camici, S., Servat, E. (2012). Estimation of antecedent wetness conditions for flood modelling in Northern Morocco. Hydrology and Earth System Sciences, 16, 4375-4386, doi:10.5194/hess-16-4375-2012.

---

## Referee Comment (RC1) · Anonymous Referee #1 · 6 Apr 2018

General comments

The paper goal is to examine the importance of soil moisture to influence the hydrological response of a catchment located in the Mediterranean (Southern France). Despite very interesting the topic is not new as many studies (also not cited by the authors) have tried to understand the role of soil moisture in flood modelling. In particular, many of them demonstrated that soil moisture is a good proxy of the catchment initial conditions and it behaves better than many API-based indexes (see the SC1 comment in the Interactive discussion). In this respect, this study has the advantage of relying upon a really dense network of soil moisture monitoring stations that can help better the

understanding of the underlying rainfall-runoff generation processes. Said that, I think that the paper is of interest for the journal readership and potentially very interesting. It is also well detailed and written. I have to major comments that the authors should take into consideration:

1) The paper is really too long and the richness of details and number despite commendable sometimes distracts from the general objectives and make the reading of the paper not really easy. I suggest to shorten the manuscript and reduce the number of figures trying to generalize a little bit the results and reducing the numbers in the text which should already evident from the figures. I think this will make the paper gaining in readability. Please, define clearly in method why certain indexes are introduced and their scope. Consider the use of a table the role of each index in case.

2) The study of the temporal/spatial variability of soil moisture and its connection with land use is surely important but at times seem disconnected with the main research question (RQ) which is the understanding of the impact of soil moisture on runoff generation. The result is that the two RQs higleted in the paper seem two distinct chapter. If the authors want to maintain such a similar structure I am convinced that connections exist and thus they should emphasized. With connections I mean the role of the land use, spatial variability and temporal variability of certain plot/station soil moisture values in the runoff generation mechanism. If this was done, it is not really immediate to get. The intro section lacks of a significant part of the literature in this topic. The authors could refer to the SC1 comment in the interactive discussion to improve this part.

Based on my comment above I recommend the paper accepted after major revisions. I would be happy to revise the paper again and to provide a more detailed technical revision once the above comments will have been addressed.

---

## Referee Comment (RC2) · H. Gao (Referee) · 9 Apr 2018

This paper reports a case study to investigate the impact of initial soil moisture on hydrological response. Although many similar experiments have been carried out in tremendous papers, I still believe that this kind of field experiments shall be encouraged for publication in hydrological journals, because experiments and observations are fundamental to test or reject our scientific hypotheses. But considering the quality of the writing, further revision is needed for consideration to publish in HESS.

1. This paper is too long with too many details. The abstract shall be shorten. Introduction looks okay, but more literature shall be discussed as mentioned by previous

two reviewers. There are too many details in the Results and Discussion. Particularly, the Conclusions have two pages, which is absolutely not necessary.

2. For the science part, the authors missed an important factor while discussing the relationship between soil moisture and hydrological response – the topography. As we all know, as hydrologists, topography has great impact on initial soil moisture and runoff generation. For example, hillslopes, riparian areas, and plateau have significantly different runoff generation mechanisms (cf. Seibert et al., 2003; Savenije, 2010; Gao et al., 2014). Figure 1 shows that many continuous soil moisture observations are located in the areas near the channel network, right? This means that the observations are mainly reflecting the soil moisture dynamic on riparian areas. This can explain the immediate response of rainfall – soil moisture – runoff. But what is happening on hillslopes? What is the impact of topography on your conclusions?

References:

Seibert, J., Rodhe, A., & Bishop, K. (2003). Simulating interactions between saturated and unsaturated storage in a conceptual runoff model. Hydrological Processes, 17(2), 379-390.

Savenije, H. H. G.: HESS Opinions "Topography driven conceptual modelling (FLEX-Topo)", Hydrol. Earth Syst. Sci., 14, 2681–2692, 2010.

Gao, H., Hrachowitz, M., Fenicia, F., Gharari, S., and Savenije, H. H. G.: Testing the realism of a topography-driven model (FLEX-Topo) in the nested catchments of the Upper Heihe, China, Hydrol. Earth Syst. Sci., 18, 1895–1915, 2014.
* * *

---

## Referee Comment (RC3) · Anonymous Referee #3 · 19 Apr 2018

The manuscript "How does initial soil moisture influence the hydrological response? A case study from southern France" by Uber et al. sets out to explore the relevance of soil moisture for the generation of floods. Although the overall topic is highly appreciated, I am not sure what the novelty of this paper is, nor did the authors convince me about what they really want to do. There are several critical points I would the authors encourage to invest a bit more effort in to develop this study:

(1) It is not entirely clear what the objective of this study is. Is it intended to give an overview of the spatial pattern of soil moisture and how these evolve over time? In spite of a relatively rich data set, is such an analysis really warranted by the data? In

other words, are the conclusions drawn generally applicable or do they merely describe the relatively local study sites? If the latter, what can be learned from that? Or is the study rather meant to improve our understanding of soil moisture to generate flows? Then much of analysis and discussion of the spatial pattern can be condensed. What I also found surprising is that in the introduction much is made of the importance of flash floods, but this is not been really picked up and discussed explicitly later with respect to the results. Why?

(2) Soil moisture and its spatio-temporal pattern have been subject to a vast body of studies in the past. In that context, neither the introduction nor the discussion of the results here do any justice to these earlier efforts. How is this study placed in the context of this earlier work? What is different? What is novel? What is the same? Could your results and interpretation improve some of the earlier interpretations? If so, how? In which aspects are the conclusions you draw similar/not similar to other studies? Why? I would argue that there is quite a lot to discover on this topic and I would strongly encourage the authors to do so to allow the reader to better appreciate the authors' efforts. In addition to the other reviewers' suggestions, I would also think that (at least) the following references are highly relevant and provide necessary context and should thus be considered: McMillan et al. (2014), McMillan and Srinivasan (2015), Hrachowitz et al. (2011) and Li et al. (2011)

(3) More information and context is needed for the soil moisture data. It would be nice to have a more detailed map of the locations of the sensors (maybe also cross-sections), to get a better idea of what is observed where. Related to that, much is made of the, admittedly quite extensive soil moisture data set. However, in section 4.1 no consideration is given to the limitations of what is actually measured. In a simplified way, the role of unsaturated water storage lies in the temporal storage of water between permanent wilting point and field capacity (i.e. water held against gravity). At any point in time, this storage is controlled by plant water use (transpiration) and soil evaporation, whereby plant water use extracts water more efficiently than soil evaporation. Thus, to

make sure to meaningfully measure soil moisture, it therefore needs to be measured exactly where plant roots extract water from the soil. Is this the case here? it was mentioned that in the vineyards the sensors were place between the vines. Is this were the most important parts of the root system of vines is to be found? I could imagine that the measurements obtained are thus largely biased towards high soil moisture, as plant water extraction may be underrepresented in these locations. How would that change the interpretation? It would be good if the authors not only acknowledged this common problem but also discussed the limitations that come with it.

(4) Throughout the manuscript, methods could be explained in a clearer and more consistent way and some of the results could be provided in a more quantitative manner. Some examples: it is stated that regression methods require the assumption of "normal distribution of dependent and independent variables". What does that mean? Why should x and y have to be normally distributed? That would only lead to clustering. Do you rather mean that the residuals need to be normally distributed? Please clarify. Another example section 3.1.1. What do you intend to say with this paragraph? Why is it important to have normally distributed soil moisture? Similarly, in section 3.1.2 it is stated that "[. . .] pdf [. . .] agree with either normal distribution or [. . .]". Please use a more formal language here and provide quantification. Do you want to say that the hypothesis that the pdf is a sample from a normal distribution cannot be rejected on a 0.0x significance level? Then please say so.

References:

Hrachowitz, M., Bohte, R., Mul, M. L., Bogaard, T. A., Savenije, H. H. G., & Uhlenbrook, S. (2011). On the value of combined event runoff and tracer analysis to improve understanding of catchment functioning in a data-scarce semi-arid area. Hydrology and Earth System Sciences, 15(6), 2007-2024

Li, H., Sivapalan, M., & Tian, F. (2012). Comparative diagnostic analysis of runoff generation processes in Oklahoma DMIP2 basins: The Blue River and the Illinois River.

[Figure]

Journal of hydrology, 418, 90-109.

McMillan, H., Gueguen, M., Grimon, E., Woods, R., Clark, M., & Rupp, D. E. (2014). Spatial variability of hydrological processes and model structure diagnostics in a 50 km2 catchment. Hydrological processes, 28(18), 4896-4913.

McMillan, H. K., & Srinivasan, M. S. (2015). Characteristics and controls of variability in soil moisture and groundwater in a headwater catchment. Hydrology and Earth System Sciences, 19(4), 1767.

---

## Author Comment (AC1) · 13 Jul 2018

Dear Mr. Brocca, Thank you for the positive evaluation of the study's setup and analyses as well as for the list of additional references on the role of initial soil moisture on the hydrologic response. We will regard the recommended literature and take these papers into account in a revised version of the article.

---

## Author Comment (AC2) · 13 Jul 2018

**Response to Anonymous Referee #1**

General comments
The paper goal is to examine the importance of soil moisture to influence the hydrological
response of a catchment located in the Mediterranean (Southern France). Despite
very interesting the topic is not new as many studies (also not cited by the authors)
have tried to understand the role of soil moisture in flood modelling. In particular, many
of them demonstrated that soil moisture is a good proxy of the catchment initial conditions
and it behaves better than many API-based indexes (see the SC1 comment in the
Interactive discussion). In this respect, this study has the advantage of relying upon
a really dense network of soil moisture monitoring stations that can help better the understanding of the
underlying rainfall-runoff generation processes. Said that, I think
that the paper is of interest for the journal readership and potentially very interesting.
It is also well detailed and written. I have to major comments that the authors should
take into consideration:

Response:

Thank you very much for your review of the manuscript. Your comments are highly appreciated and will
help us to improve the current version of the manuscript.

Thank you for the positive evaluation of the topic of the study and the appreciation of the data set.
Regarding the missing literature references, we want to stress that our study is not a modelling exercise.
Thus, in the literature review, we chose to focus on papers that rely on the analysis of soil moisture,
discharge and precipitation measurements as does our study. But of course flood modelling studies are
undisputedly very insightful to address the topic of the paper. We will reconsider the existing literature
on the role of soil moisture on the hydrological response. The short comment posted by Mr. Brocca will
be very useful.

1) The paper is really too long and the richness of details and number despite commendable
sometimes distracts from the general objectives and make the reading of the
paper not really easy. I suggest to shorten the manuscript and reduce the number of
figures trying to generalize a little bit the results and reducing the numbers in the text
which should already evident from the figures. I think this will make the paper gaining
in readability. Please, define clearly in method why certain indexes are introduced and
their scope. Consider the use of a table the role of each index in case.

This was also brought up by Referee #2 (H. Gao). We agree that the manuscript can be shortened and we
will be more concise for both the text and the Figures in the revised version. We suggest to omit figures
no. 2, 12 and 13 at least and possibly figures 5 and/or 7. Figure 4 could be integrated into figure 3. The
methods section 2.4 will be revised and we will consider using a table to better justify the indices.

2) The study of the temporal/spatial variability of soil moisture and its connection with
land use is surely important but at times seem disconnected with the main research
question (RQ) which is the understanding of the impact of soil moisture on runoff generation.
The result is that the two RQs higleted in the paper seem two distinct chapter.
If the authors want to maintain such a similar structure I am convinced that connections
exist and thus they should emphasized. With connections I mean the role of the

land use, spatial variability and temporal variability of certain plot/station soil moisture values in the runoff generation mechanism. If this was done, it is not really immediate to get. The intro section lacks of a significant part of the literature in this topic. The authors could refer to the SC1 comment in the interactive discussion to improve this part.

Thank you very much for this valuable comment. We regret that we could not convey the connections between the two research questions well enough. In a revised version of the paper we will try to emphasize that the first objective to study the variability of soil moisture is necessary to obtain a relevant and representative value for soil moisture at the catchment scale. This is crucial to address the main research question of the role of initial soil moisture on the hydrologic response of a catchment.

Based on my comment above I recommend the paper accepted after major revisions. I would be happy to revise the paper again and to provide a more detailed technical revision once the above comments will have been addressed.

Thank you for this recommendation and for your offer to provide a second revision of a reworked manuscript.

---

## Author Comment (AC3) · 13 Jul 2018

**Answer to H. Gao (Referee #2)**

H.Gao:

This paper reports a case study to investigate the impact of initial soil moisture on hydrological response. Although many similar experiments have been carried out in tremendous papers, I still believe that this kind of field experiments shall be encouraged for publication in hydrological journals, because experiments and observations are fundamental to test or reject our scientific hypotheses. But considering the quality of the writing, further revision is needed for consideration to publish in HESS.

Response:

We want to thank you for your review of the manuscript and for appreciating the usefulness of our study. Considering your comments and also those of the other Reviewers, we agree to revise the manuscript according to the comments given.

1. This paper is too long with too many details. The abstract shall be shorten. Introduction looks okay, but more literature shall be discussed as mentioned by previous two Reviewers. There are too many details in the Results and Discussion. Particularly, the Conclusions have two pages, which is absolutely not necessary.

We agree that the manuscript can be shortened. We will shorten these sections as proposed. Also, as recommended by Reviewer 1, we will omit figures 2,12 and 13, possibly also 5 and 7, and try to combine figures 3 and 4.

We will consider the literature proposed by Reviewer 1 and L. Brocca, the author of short comment 1. Thank you for the further literature recommendations that you provided. We will also refer to these articles.

2. For the science part, the authors missed an important factor while discussing the relationship between soil moisture and hydrological response – the topography. As we all know, as hydrologists, topography has great impact on initial soil moisture and runoff generation. For example, hillslopes, riparian areas, and plateau have significantly different runoff generation mechanisms (cf. Seibert et al., 2003; Savenije, 2010; Gao et al., 2014). Figure 1 shows that many continuous soil moisture observations are located in the areas near the channel network, right? This means that the observations are mainly reflecting the soil moisture dynamic on riparian areas. This can explain the immediate response of rainfall – soil moisture – runoff. But what is happening on hillslopes? What is the impact of topography on your conclusions?

We agree that topography plays an important role in the relationship of soil moisture and the hydrologic response. It is true that we only mention it among other factors that determine runoff generation. In our study we decided to focus on land use rather than on topography as our initial hypothesis was that –in our catchments - land use and the associated different soil types determine this relationship more than other factors such as topography and geology etc.

Following your comment, we looked into the relationship between topography and initial soil moisture. There seems to be no direct relationship between the distance to the river and the plot mean soil moisture. That is illustrated by the following figure where every point represents the plot mean of initial soil moisture (the different colors of the points represent the different events).

[Figure]

We agree that it is not ideal that the locations of the measurements are clustered in the south of the Gazel catchment close to the stream and not evenly distributed in the catchment. This is due to practical reasons (e.g. protected sites for the installation of the probes available at Le Pradel, linked with the urgency of on-alert measurements that had to be completed before the rain started, which was always challenging). The distance of the continuous soil moisture measurement sites at le Pradel is > 40 m for all sites. As the stream is incised into the bedrock, all measurement sites are at a considerably higher elevation than the stream bed. They do represent hillslopes that are not connected to the river via influent groundwater. Only two of the three sensors installed in the north of the catchments are located in grasslands that might be connected to the river during rain events. Concerning the three landscape units that are defined by topography in the reference you provided (Savenije 2010), measurement sites represent these units very well as there are measurement sites in riparian areas, on hillslopes and on plateaus. We will include this information in a revised version of the manuscript.

We agree, that this is not evident from figure 1, which is misleading in this regard and we thank you for pointing this out. We will improve figure 1 by providing a closer zoom into the area in the south of the Gazel catchment, inserting a scale bar in the zoom and by providing information on the land use of the sites of the measurement sites. Your comment is in agreement with the comment of Reviewer 3 who also demands a more detailed map, so we will provide a revised figure which hopefully helps to better characterize the measurement sites.

---

## Author Comment (AC4) · 13 Jul 2018

**Response to Anonymous Referee #3**

Anonymous Referee #3:

The manuscript "How does initial soil moisture influence the hydrological response? A case study from southern France" by Uber et al. sets out to explore the relevance of soil moisture for the generation of floods. Although the overall topic is highly appreciated, I am not sure what the novelty of this paper is, nor did the authors convince me about what they really want to do. There are several critical points I would the authors encourage to invest a bit more effort in to develop this study:

Response:

Thank you very much for the review of our paper. Your comments are highly appreciated and will help us to improve the manuscript. We regret that we could not convey the novelty of the paper sufficiently. Thank you for pointing out this important fact. In the revised version of the paper we will emphasize more clearly the novelty of the paper which lies on:

- Obtaining meaningful estimates of soil moisture at the catchment scale from a dense network of in-situ measurements that were derived from two different measurement schemes without relying on satellite data, modeled soil moisture, antecedent precipitation etc.
- Using this information to analyze the relation between initial soil moisture and the event based runoff coefficient which was equally obtained from using high resolution measurement data. Its uncertainty due to stage discharge transformation and hydrograph separation was scrutinized.

To our knowledge, no previous studies addressed this question with a similar data basis and for as many events.

(1) It is not entirely clear what the objective of this study is. Is it intended to give an overview of the spatial pattern of soil moisture and how these evolve over time? In spite of a relatively rich data set, is such an analysis really warranted by the data? In other words, are the conclusions drawn generally applicable or do they merely describe the relatively local study sites? If the latter, what can be learned from that? Or is the study rather meant to improve our understanding of soil moisture to generate flows? Then much of analysis and discussion of the spatial pattern can be condensed.

The general objective of the study is the latter: improving our understanding of the role of soil moisture in generating runoff that leads to flash floods. However, it is not easy to obtain a meaningful measure of catchment mean soil moisture due to 3-D spatial variability across scales and temporal changes within hours. To obtain this measure, we believe it is important to regard the spatial variability across scales. We agree that this analysis and the results regarding spatial patterns can be condensed. We will shorten these sections in the revised version of the manuscript and make our objectives clearer.

What I also found surprising is that in the introduction much is made of the importance of flash floods, but this is not been really picked up and discussed explicitly later with respect to the results. Why?

Thank you for pointing that out. The importance of flash floods is the motivation of our study but we do not study flash floods in this manuscript, so we will shorten that aspect in the introduction.

(2) Soil moisture and its spatio-temporal pattern have been subject to a vast body of

studies in the past. In that context, neither the introduction nor the discussion of the results here do any justice to these earlier efforts. How is this study placed in the context of this earlier work? What is different? What is novel? What is the same? Could your results and interpretation improve some of the earlier interpretations? If so, how? In which aspects are the conclusions you draw similar/not similar to other studies? Why? I would argue that there is quite a lot to discover on this topic and I would strongly encourage the authors to do so to allow the reader to better appreciate the authors' efforts. In addition to the other Reviewers' suggestions, I would also think that (at least) the following references are highly relevant and provide necessary context and should thus be considered: McMillan et al. (2014), McMillan and Srinivasan (2015), Hrachowitz et al. (2011) and Li et al. (2011)

Thank you for the additional literature recommendations which will be valuable for the revised version of the manuscript. We will re-examine the literature using these recommendations as well as the ones of the other Reviewers and better link it to our results and conclusions.

(3) More information and context is needed for the soil moisture data. It would be nice to have a more detailed map of the locations of the sensors (maybe also crosssections), to get a better idea of what is observed where.

Improvement of figure 1 was also demanded by Reviewer 2. We will provide a revised figure with a closer zoom into the area in the south of the Gazel catchment. We will provide information on the land use of the location of the measurement sites. Our experimental plan was not organized with a real upstream-downstream geometry which makes, in our opinion, cross-sections not relevant to this particular study. However, Fig. 1 caption will be modified to give precisions about the sensors depths, additionally to the main text so that the vertical profile of the measurements is better conveyed. (Just as a reminder: continuous measurements probes were installed at different depths at one location per plot, the on-alert measurements were conducted at approx. 10 randomly chosen points per field).

Related to that, much is made of the, admittedly quite extensive soil moisture data set. However, in section 4.1 no consideration is given to the limitations of what is actually measured. In a simplified way, the role of unsaturated water storage lies in the temporal storage of water between permanent wilting point and field capacity (i.e. water held against gravity). At any point in time, this storage is controlled by plant water use (transpiration) and soil evaporation, whereby plant water use extracts water more efficiently than soil evaporation. Thus, to make sure to meaningfully measure soil moisture, it therefore needs to be measured exactly where plant roots extract water from the soil. Is this the case here? it was mentioned that in the vineyards the sensors were place between the vines. Is this were the most important parts of the root system of vines is to be found? I could imagine that the measurements obtained are thus largely biased towards high soil moisture, as plant water extraction may be underrepresented in these locations. How would that change the interpretation? It would be good if the authors not only acknowledged this common problem but also discussed the limitations that come with it.

We agree that transpiration is crucial in determining initial soil moisture. The locations of the sensors in the vineyards where carefully chosen to represent a compromise between feasibility and representativeness. The root system of the vines can reach up to 4 m depth, so it was impossible to measure soil moisture throughout the root system. By installing the permanent sensors in the vine rows in a depth of up to 50 cm the sensors are as close as possible to the root system. Considering a vineyard

plot that consists of the vine rows and the (nearly) bare soil between the rows, we do not believe that we underestimate plot-scale soil moisture due to transpiration.

The manual measurements in the topsoil were conducted between the rows, because this is where surface runoff started (Visual inspection during heavy rain). We also want to stress here that measuring soil moisture in vineyards with clayey and extremely stony soils is very challenging. To insert the 6 cm rods of the manual soil moisture probes in the stony soil without damaging the instrument, we often had to try more than 10 times before we managed to completely insert the rods.

We agree that this justification is not conveyed in the section where the measurements are described and will add an explanation for the location of the sensors and account for the problem of soil moisture heterogeneity due to transpiration in section 2.2.

(4) Throughout the manuscript, methods could be explained in a clearer and more consistent way

We regret that the methods were not explained clearly and consistently. As proposed by Reviewer 1 we will consider to use a table to explain the different variables calculated in section 2.4. We hope that will make it clearer and will allow us to shorten this section as you proposed in (1).

[…] and some of the results could be provided in a more quantitative manner. Some examples: it is stated that regression methods require the assumption of "normal distribution of dependent and independent variables". What does that mean? Why should x and y have to be normally distributed? That would only lead to clustering. Do you rather mean that the residuals need to be normally distributed? Please clarify.

Thank you for pointing that out. In order to shorten the manuscript we decided to delete the sentence you are referring to as this sentence is indeed not very clear and not necessary for the paper.

Another example section 3.1.1. What do you intend to say with this paragraph? Why is it important to have normally distributed soil moisture?

This paragraph is aimed to describe soil moisture at the plot scale, which is important for the first objective, to obtain a meaningful method for the catchment mean value. The distribution of the values helps interpreting the significance of the mean since an exhaustive sample related to all factors (land use, geology, slope, …) could not be obtained. Thus, only a statistical approach can be used to assess its uncertainty. As you proposed in your comment (1) we will shorten this section and the following.

Similarly, in section 3.1.2 it is stated that "[: : :] pdf [: : :] agree with either normal distribution or [: : :]". Please use a more formal language here and provide quantification. Do you want to say that the hypothesis that the pdf is a sample from a normal distribution cannot be rejected on a 0.0x significance level? Then please say so.

We agree to reformulate that. Thank you for pointing it out.

---

## Author Response (AR1)

**Point-by-point response to the Referee Reports**

This document is formatted as follows:

- 5 comments from the referees are shown in blue
  - author's responses are shown in black
  - author's changes in the manuscript are shown in italics
    - 1. Response to Anonymous Referee #1:
- 10

**General comments**

The paper goal is to examine the importance of soil moisture to influence the hydrological response of a catchment located in the Mediterranean (Southern France). Despite

- 15 very interesting the topic is not new as many studies (also not cited by the authors) have tried to understand the role of soil moisture in flood modelling. In particular, many of them demonstrated that soil moisture is a good proxy of the catchment initial conditions and it behaves better than many API-based indexes (see the SC1 comment in the Interactive discussion). In this respect, this study has the advantage of relying upon
- 20 a really dense network of soil moisture monitoring stations that can help better the understanding of the underlying rainfall-runoff generation processes. Said that, I think that the paper is of interest for the journal readership and potentially very interesting. It is also well detailed and written. I have to major comments that the authors should take into consideration:
- 25

Thank you very much for your review of the manuscript. Your comments are highly appreciated and will helped us to improve the the manuscript. Thank you for the positive evaluation of the topic of the study and the appreciation of the data set.

Regarding the missing literature references, we want to stress that our study is not a modelling exercise.

- 30 Thus, in the literature review, we chose to focus on papers that rely on the analysis of soil moisture, discharge and precipitation measurements as does our study. But of course flood modelling studies are undisputedly very insightful to address the topic of the paper. We re-examined the literature on the subject and added the references listed below under "List of additional literature" (recommended by the reviewers and additional ones) and referred to them in the Introduction and Discussion section.
- 35 Concerning the finding of several studies that soil moisture is an asset for the estimation of catchment initial conditions we added the sentence "Numerous modelling studies have shown the high sensitivity of the modelled runoff response to dini and the importance of estimates of dini at the catchment scale as initial conditions in event-based models (e.g. Castillo et al., 2003; Huang et al., 2007; LeLay and Saulnier, 2007; Berthet et al., 2009; Brocca et al., 2009b; Tramblay et al., 2010, 2012; Li et al., 2012; Massari et al., 2010, 2012; Li et al., 2012; Massari et al., 2010, 2012; Li et al., 2012; Massari et al., 2010, 2012; Li et al., 2012; Massari et al., 2010, 2012; Li et al., 2012; Massari et al., 2010, 2012; Li et al., 2012; Massari et al., 2010, 2012; Li et al., 2012; Massari et al., 2010, 2012; Li et al., 2012; Massari et al., 2010, 2012; Li et al., 2012; Massari et al., 2010, 2012; Li et al., 2012; Massari et al., 2010, 2012; Li et al., 2012; Massari et al., 2010, 2012; Li et al., 2012; Massari et al., 2010, 2012; Li et al., 2012; Massari et al., 2010, 2012; Li et al., 2012; Massari et al., 2010, 2012; Li et al., 2012; Massari et al., 2010, 2012; Li et al., 2012; Massari et al., 2010, 2012; Li et al., 2012; Massari et al., 2010, 2012; Li et al., 2012; Massari et al., 2010, 2012; Li et al., 2012; Massari et al., 2010, 2012; Li et al., 2012; Massari et al., 2010, 2012; Li et al., 2012; Massari et al., 2010, 2012; Li et al., 2010; Massari et al., 2010; Massa
- 40 2014a,b, 2015; Grillakis et al., 2016)." (p.3, l.34).

1) The paper is really too long and the richness of details and number despite commendable sometimes distracts from the general objectives and make the reading of the paper not really easy. I suggest to shorten the manuscript and reduce the number of

45 figures trying to generalize a little bit the results and reducing the numbers in the text which should already evident from the figures. I think this will make the paper gaining

in readability. Please, define clearly in method why certain indexes are introduced and their scope. Consider the use of a table the role of each index in case.

This was also brought up by Referee #2 (H. Gao). We agree that the manuscript was too long and
shortened the paper substantially. The main text was shortened by three pages. We deleted figures 2,12 and 13 of the original manuscript and figure 4 was integrated into figure 3. To do so, two subfigures of figure 3 were deleted. Several numbers were deleted in the text in the results section (p.14, l. 10, l.15 and l. 16; p. 15 l. 8-10 and l. 16; p. 16 l.20). As you proposed we included a table (Table 1 in the new manuscript) that gives the role of each index and helps to describe the soil moisture data analysis more

10 concisely.

2) The study of the temporal/spatial variability of soil moisture and its connection with land use is surely important but at times seem disconnected with the main research question (RQ) which is the understanding of the impact of soil moisture on runoff generation.

15 The result is that the two RQs higleted in the paper seem two distinct chapter. If the authors want to maintain such a similar structure I am convinced that connections exist and thus they should emphasized. With connections I mean the role of the land use, spatial variability and temporal variability of certain plot/station soil moisture

values in the runoff generation mechanism. If this was done, it is not really immediate

20 to get. The intro section lacks of a significant part of the literature in this topic. The authors could refer to the SC1 comment in the interactive discussion to improve this part.

Thank you very much for this valuable comment. We regret that we could not convey the connections

- 25 between the two research questions well enough. In the revised version much of the analysis of the impact of land use on soil moisture was shortened. In this way we hope that the impression of two distinct chapters is diminished. We emphasize that the first objective (to study the variability of soil moisture) is necessary to obtain a relevant and representative value for soil moisture at the catchment scale. This is crucial to address the main research question of the role of initial soil moisture on the
- 30 hydrologic response of a catchment. Thus we rephrased the description of the objectives in the abstract: "Thus, the objectives are twofold: (1) obtaining meaningful estimates of soil moisture at catchment scale from a dense network of in-situ measurements and (2) using this estimate of  $\mathscr{T}_{ini}$  to analyze its relation with  $\phi ev$  calculated for many runoff events."(p.1 l.19)

Further, in the introduction we no longer formulate our objectives as 2 research questions but rephrased

35 it: "Relying solely on in-situ data, we aim to (i) obtain a meaningful estimate of catchment scale soil moisture and its uncertainty and (ii) answer the research question how does soil moisture at the event onset affect the hydrological response?" (p.5 l.1)

This was done in the conclusion as well by rephrasing the introductory sentence to: "To this end, two issues were addressed and exemplarily examined in the nested Gazel (3.4 km2) and Claduègne (43 km2)

- 40 catchments: (1) Obtaining a meaningful estimate of soil moisture at the catchment scale and (2) Analyzing the relation between initial soil moisture  $\tilde{\theta}_{ini}$  and the hydrological response quantified as the event-based runoff coefficient  $\phi_{ev}$ ." (p.21 I.29)
  - Based on my comment above I recommend the paper accepted after major revisions.
- 45 I would be happy to revise the paper again and to provide a more detailed technical

revision once the above comments will have been addressed.

Thank you for this recommendation and for your offer to provide a second revision of a reworked manuscript.

5

2. Response to H. Gao (Referee 2)

This paper reports a case study to investigate the impact of initial soil moisture on hydrological response. Although many similar experiments have been carried out in

10 tremendous papers, I still believe that this kind of field experiments shall be encouraged for publication in hydrological journals, because experiments and observations are fundamental to test or reject our scientific hypotheses. But considering the quality of the writing, further revision is needed for consideration to publish in HESS.

We want to thank you for your review of the manuscript and for appreciating the usefulness of ourstudy. We revised the manuscript according to your comments and the ones of the other reviewers.

1. This paper is too long with too many details. The abstract shall be shorten. Introduction looks okay, but more literature shall be discussed as mentioned by previous two reviewers. There are too many details in the Results and Discussion. Particularly, the Conclusions have two pages, which is absolutely not necessary.

20 We shortened the manuscript as proposed esp. the sections you pointed out. Also, as recommended by reviewer 1, we deleted figures 2,12 and 13 of the original manuscript and figure 4 was integrated into figure 3.

Thank you for the further literature recommendations that you provided.

We re-examined the literature and added the references listed below under "List of additional literature"
 (recommended by the reviewers and additional ones) and referred to them in the Introduction and Discussion section.

2. For the science part, the authors missed an important factor while discussing the relationship between soil moisture and hydrological response – the topography. As we all know, as hydrologists, topography has great impact on initial soil moisture and runoff

- 30 generation. For example, hillslopes, riparian areas, and plateau have significantly different runoff generation mechanisms (cf. Seibert et al., 2003; Savenije, 2010; Gao et al., 2014). Figure 1 shows that many continuous soil moisture observations are located in the areas near the channel network, right? This means that the observations are mainly reflecting the soil moisture dynamic on riparian areas. This can explain
- 35 the immediate response of rainfall soil moisture runoff. But what is happening on hillslopes? What is the impact of topography on your conclusions?

We agree that topography plays an important role in the relationship of soil moisture and the hydrologic response. It is true that we only mention it among other factors that determine runoff generation. In our study we desided to factor an land use rather than on topography as our initial hyperbasis was that in

40 study we decided to focus on land use rather than on topography as our initial hypothesis was that –in

our catchments - land use and the associated different soil types determine this relationship more than other factors such as topography and geology etc.

Following your comment, we looked into the relationship between topography and initial soil moisture.
There seems to be no direct relationship between the distance to the river and the plot mean soil moisture. That is illustrated by the following figure where every point represents the plot mean of initial soil moisture (the different colors of the points represent the different events).

Distance to stream [m]

We agree that it is not ideal that the locations of the measurements are clustered in the south of the

- 10 Gazel catchment close to the stream and not evenly distributed in the catchment. This is due to practical reasons (e.g. protected sites for the installation of the probes available at Le Pradel, linked with the urgency of on-alert measurements that had to be completed before the rain started, which was always challenging). The distance of the continuous soil moisture measurement sites at le Pradel is > 40 m for all sites. As the stream is incised into the bedrock, all measurement sites are at a considerably higher
- 15 elevation than the stream bed. They do represent hillslopes that are not connected to the river via influent groundwater. Only two of the three sensors installed in the north of the catchments are located in grasslands that are potentially connected to the river during rain events.

We added a sentence in the Section 2.2 Data availability to give information on topography: "Concerning
 topography, most of the sensors are located on hillslopes which is the dominant topographic zone
 according to Savenije (2010) in the catchment. Only two plots are located in the riparian area and are
 potentially connected to the stream during rain events." (p.6 l. 1)

We agree, that the information on distance to the river is not evident from figure 1, which is misleading in this regard and we thank you for pointing this out. Your comment is in agreement with the comment of Reviewer 3 who also demands a more detailed map. We revised figure 1 and show a closer zoom into the area in the south of the Gazel catchment, a scale bar in the zoom window and by provide information on the land use of the sites of the measurement sites.

**5**

**3. Response to Anonymous Referee #3:**

The manuscript "How does initial soil moisture influence the hydrological response? A case study from southern France" by Uber et al. sets out to explore the relevance of

- 10 soil moisture for the generation of floods. Although the overall topic is highly appreciated, I am not sure what the novelty of this paper is, nor did the authors convince me about what they really want to do. There are several critical points I would the authors encourage to invest a bit more effort in to develop this study:
- 15 Thank you very much for the review of our paper. Your comments are highly appreciated and helped us to improve the manuscript. We regret that we could not convey the objectives and the novelty of the paper sufficiently. Thank you for pointing out this important fact. In the revised version of the paper we reformulated the objectives and stress the novelty of the paper in the last sentences of the introduction: *"Thus, this study's novelty is to analyze the relation between*  $\phi_{ev}$  and  $\vartheta_{ini}$  when both are obtained from a
- 20 comprehensive, high resolution data set allowing the assessment of the uncertainty of the two variables. Relying solely on in-situ data, we aim to (i) obtain a meaningful estimate of catchment scale soil moisture and its uncertainty and (ii) answer the research question how does soil moisture at the event onset affect the hydrological response?" (p. 4 I. 33)

(1) It is not entirely clear what the objective of this study is. Is it intended to give an

- 25 overview of the spatial pattern of soil moisture and how these evolve over time? In spite of a relatively rich data set, is such an analysis really warranted by the data? In other words, are the conclusions drawn generally applicable or do they merely describe the relatively local study sites? If the latter, what can be learned from that? Or is the study rather meant to improve our understanding of soil moisture to generate flows?
- 30 Then much of analysis and discussion of the spatial pattern can be condensed.

As suggested, we shortened large parts of the analysis of the spatial patterns and the influence on land use of soil moisture (Methods: section 2.4 was considerably shortened; Results: sections 3.1.1, 3.1.2 and 3.2 in the original manuscript are combined in the much shorter sect. 3.1 in the new manuscript;

35 Discussion: section 4.1 and 4.2 are combined in the shorter sect. 4.1 of the new manuscript).

Furthermore, we reformulated our objectives. Please see our response to comment 2) of referee 1 for the changes made in the manuscript to reformulate the objectives.

- 40 In this way we focus much more concisely on the objectives of (i) obtaining meaningful estimates of soil moisture at catchment scale as well as its uncertainty and (ii) using this information to analyze the relation between initial soil moisture and the hydrological response. The latter is quantified with the event based runoff coefficient while also considering its uncertainty.
- 45 What I also found surprising is that in the introduction much is made of the importance of flash

floods, but this is not been really picked up and discussed explicitly later with respect to the results. Why?

Thank you for pointing that out. The importance of flash floods is the motivation of our study but we did

5 not study flash floods in this manuscript. That aspect was shortened in the introduction (p.2 l.9 and l.12, l. 19-27).

In the discussion the potential of our results for threshold based flash flood warning systems as proposed by other authors is picked up (p.21 l.1 -14).

- 10 (2) Soil moisture and its spatio-temporal pattern have been subject to a vast body of studies in the past. In that context, neither the introduction nor the discussion of the results here do any justice to these earlier efforts. How is this study placed in the context of this earlier work? What is different? What is novel? What is the same?
- 15 We added the references listed below under "List of additional literature". In the revised version of the manuscript, we stress the novelty of the study (see above). Our studies is different to many studies that address this topic using indirect data or model output. In order to stress this point, we added the sentences "*These approaches [indirect information] offer many advantages, such as the global availability of remote sensing data and the easier acquisition of this data (e.g. Brocca et al., 2009c).*
- 20 Numerous studies found good agreement of soil moisture data obtained from in-situ measurements and remote sensing (e.g. Brocca et al., 2009c, 2013, Huza et al., 2014) Nonetheless, case studies are important to confirm the results obtained with indirect data as well as the results from modelling exercises and to thoroughly understand the hydrologic functioning of local sites." (p.4, l.26) Further, we added term "Relying solely on in-situ data, we [...]" (p.5, l.1) in the last sentence of the introduction that sums up our objectives.

Could your results and interpretation improve some of the earlier interpretations? If so, how? In which aspects are the conclusions you draw similar/not similar to other studies? Why? I would argue that there is quite a lot to discover on this topic and I

30 would strongly encourage the authors to do so to allow the reader to better appreciate the authors' efforts.

In the discussion, we added the following phrase to compare our results concerning the maximum number of samples needed to estimate mean soil moisture to other studies results: "This is consistent

- 35 with the results of Zucco et al. (2014) who found a maximum number of 11 or 20 required samples at the plot scale and catchment scale respectively and the ones of Molina et al. (2014) who concluded that plot mean soil moisture in a Mediterranean mountain area was well represented with 9 probes. The review by Vereecken et al. (2014) shows that there is a wide range of estimates for these numbers and that they are site specific." (p. 17, 1.11)
- 40

Concerning the threshold observed in the relationship between initial soil moisture and event based runoff coefficients we rephrased the original sentence and included the literature you recommended: "Threshold effects in the relation of  $\phi_{ev}$  and  $\tilde{\theta}_{ini}$  are also observed by other authors (e.g. McMillan et al., 2014; Hrachowitz et al., 2011). In the Mediterranean context, the thresholds obtained by Huza et al.

(2014) in the Gazel catchment and Braud et al. (2014) in the Valescure catchment (22 and 25 vol% respectively) are lower than the one obtained here. The threshold at 45 vol% observed by Penna et al. (2011) in a 1.9 km2 headwater catchment in the Italian Dolomites on the other hand is higher than the one obtained here. McMillan et al. (2014) show that thresholds in different subcatchments of a 50 km2

catchment in New Zealand are highly variable: They range between 27 and 58 vol% and are more or less pronounced in different subcatchments." (p. 20, 1.7)

The conclusions that we draw from our work are similar to the ones of other authors. Our thorough
analysis of the uncertainties inherent in the estimation of mean catchment soil moisture and the event
based runoff coefficient further supports the conclusion that including soil moisture thresholds in
operational flood warning systems is not applicable yet (p.20 l. 1 - 18). We suggest further research to
advance on this problem by adding the phrase *"Further research and instrumentation could include the installation of piezometers in the catchment to understand subsurficial flow in the catchment, using*

10 tracers other than EC to differentiate subsurficial stormflow as a third flow component during hydrograph separation as well as the application of multivariate regression analysis methods that systematically examine different controls on  $\phi_{ev}$  such as meteorological forcing as well as  $\tilde{\theta}_{ini}$  and their interactions".

In addition to the other Reviewers' suggestions, I would also think

15 that (at least) the following references are highly relevant and provide necessary context and should thus be considered: McMillan et al. (2014), McMillan and Srinivasan (2015), Hrachowitz et al. (2011) and Li et al. (2011)

Thank you for the additional literature recommendations. They were included in the revised manuscript.

20

3) More information and context is needed for the soil moisture data. It would be nice to have a more detailed map of the locations of the sensors (maybe also crosssections), to get a better idea of what is observed where.

25 Information on topography of the soil moisture measurement sites was added: "Concerning topography, most of the sensors are located on hillslopes which is the dominant topographic zone according to Savenije (2010) in the catchment. Only two plots are located in the riparian area and are potentially connected to the stream during rain events." (p.5, l.1). And "All of these sites [on-alert measurement sites] are located on hillslopes." (p.5, l. 13).

30

Improvement of figure 1 was also demanded by Reviewer 2. We will provide a revised figure with a closer zoom into the area in the south of the Gazel catchment. We will provide information on the land use of the location of the measurement sites. Our experimental plan was not organized with a real upstream-downstream geometry which makes, in our opinion, cross-sections not relevant to this

- 35 particular study. However, Fig. 1 caption will be modified to give precisions about the sensors depths, additionally to the main text so that the vertical profile of the measurements is better conveyed. (Just as a reminder: continuous measurements probes were installed at different depths at one location per plot, the on-alert measurements were conducted at approx. 10 randomly chosen points per field).
- 40 We revised figure 1 and show a closer zoom into the area in the south of the Gazel catchment, a scale bar in the zoom window and by provide information on the land use of the sites of the measurement sites.
- Our experimental plan was not organized with a real upstream-downstream geometry which makes, in our opinion, cross-sections not relevant to this particular study. However, Fig. 1 caption was modified to give precisions about the sensors depths, additionally to the main text so that the vertical profile of the measurements is better conveyed. (Just as a reminder: continuous measurements probes were installed at different depths at one location per plot, the on-alert measurements were conducted at approx. 10 randomly chosen points per field).

Related to that, much is made of the, admittedly quite extensive soil moisture data set. However, in section 4.1 no consideration is given to the limitations of what is actually measured. In a simplified way, the role of unsaturated water storage lies in the temporal storage of water between

- 5 permanent wilting point and field capacity (i.e. water held against gravity). At any point in time, this storage is controlled by plant water use (transpiration) and soil evaporation, whereby plant water use extracts water more efficiently than soil evaporation. Thus, to make sure to meaningfully measure soil moisture, it therefore needs to be measured exactly where plant roots extract water from the soil. Is this the case here? it was
- 10 mentioned that in the vineyards the sensors were place between the vines. Is this were the most important parts of the root system of vines is to be found? I could imagine that the measurements obtained are thus largely biased towards high soil moisture, as plant water extraction may be underrepresented in these locations. How would that change the interpretation? It would be good if the authors not only acknowledged this
- 15 common problem but also discussed the limitations that come with it.

We agree that transpiration is crucial in determining initial soil moisture. The locations of the sensors in the vineyards where carefully chosen to represent a compromise between feasibility and representativeness. The root system of the vines can reach up to 4 m depth, so it was impossible to

- 20 measure soil moisture throughout the root system. By installing the permanent sensors in the vine rows in a depth of up to 50 cm the sensors are as close as possible to the root system. Considering a vineyard plot that consists of the vine rows and the (nearly) bare soil between the rows, we do not believe that we underestimate plot-scale soil moisture due to transpiration.
- The manual measurements in the topsoil were conducted between the rows, because this is where surface runoff started (Visual inspection during heavy rain). We also want to stress here that measuring soil moisture in vineyards with clayey and extremely stony soils is very challenging. To insert the 6 cm rods of the manual soil moisture probes in the stony soil without damaging the instrument, we often had to try more than 10 times before we managed to completely insert the rods.
- 30 We agree that this justification was not conveyed in the original version of the manuscript and rephrased the sentence on the location of the sensors in the vineyard as follows: "*The sensors in the vineyards were installed between two vine plants in a row which is a compromise between feasibility and representativeness of soil moisture in the vineyards which is heterogeneous due to transpiration.*" (p.5, I.5) To better justify the location of on-alert measurements in the vineyards we rephrased the sentence:
- 35 "In the vineyards the measurements were conducted in between the rows of vine plants because this is where surface runoff started (visual inspection)." (p.6, l.17)

**(4) Throughout the manuscript, methods could be explained in a clearer and more consistent way**

40

We regret that the methods were not explained clearly and consistently. As proposed by Reviewer 1 we inserted a table to explain the different variables calculated in section 2.4 (Table 1 of the new manuscript).

45 [...] and some of the results could be provided in a more quantitative manner. Some examples: it is stated that regression methods require the assumption of "normal distribution of dependent and independent variables". What does that mean? Why should x and y have to be normally distributed? That would only lead to clustering. Do you rather mean that the residuals need to be normally distributed? Please clarify.

Thank you for pointing that out. In order to shorten the manuscript we decided to delete the sentence you are referring to as this sentence is indeed not very clear and not necessary for the paper.

5 Another example section 3.1.1. What do you intend to say with this paragraph? Why is it important to have normally distributed soil moisture?

This paragraph is aimed to describe soil moisture at the plot scale, which is important for the first objective, to obtain a meaningful method for the catchment mean value. The distribution of the values helps interpreting the significance of the mean since an exhaustive sample related to all factors (land use, geology, slope, ...) could not be obtained. Thus, only a statistical approach can be used to assess its uncertainty. As you proposed in your comment (1) we will shortened this section and the following.

Similarly, in section 3.1.2 it

10

30

- 15 is stated that "[:::] pdf [:::] agree with either normal distribution or [:::]". Please use a more formal language here and provide quantification. Do you want to say that the hypothesis that the pdf is a sample from a normal distribution cannot be rejected on a 0.0x significance level? Then please say so.
- 20 Thank you for pointing that out. The respective sentence was also deleted to shorten the manuscript.
  - 4. List of additional literature

[revised manuscript text omitted]
_{\text{ev}}) = \frac{1}{n_{\text{f}}} \sum_{i=1}^{n_{\text{f}}} \theta\left(x_{i,j}, t_{\text{ev}}\right) \tag{1}$$

$$= \frac{1}{\overline{\theta}_{\mathrm{lu}}(t_{\mathrm{ev}})} = \frac{1}{n_{j_{\mathrm{tur}}}} \sum_{j_{\mathrm{lur}}=1}^{n_{j_{\mathrm{tur}}}} \overline{\theta}_{j}(t_{\mathrm{ev}}); \qquad j_{\mathrm{lur}} \in \mathrm{lu}; \mathrm{lu} \in c_{\mathrm{lur}}$$
(2)

with  $n_{\text{thr}}$  = number of plots in the respective land use class.

Catchment mean values  $\overline{\overline{\theta}}_{ev}$  are computed as a mean of the different land use classes weighted with the number of measurements per land use class:

$$\overline{\overline{\theta}}_{ev} = \frac{1}{n_{\overline{p}}} \sum_{u}^{n_{\overline{e}_{\overline{h}\overline{u}}}} \overline{\overline{\theta}}_{u} \cdot n_{\overline{h}u}, \quad u \in c_{u}$$
(3)

with  $n_{c_{tw}}$  = number of land use classes  $(n_{c_{tw}} = 4)$  and  $n_p$  = total number of plots  $(n_p = \sum n_{j_{tw}} = 11)$ .

15

5 Plot means and catchment means obtained from for the continuously measured data are computed in the same way for all three

layers  $I_{\underline{\cdot}}$  (therefore denoted  $\overline{\theta}_{ev,1}$ ) with the exception, that Here, the plot mean is obtained by averaging not only the probes installed at the same depth and the same location, but also several-all measurements in a dry period of two hours before the onset or after the end of the rain event in order to diminish noise. The catchment mean averaged over the three layers  $\tilde{\theta}_{ev}$ , i.e. considering the topmost 60 cm, is calculated from the continuously measured data (Eq. 4, Table XX1).

10 The profile mean  $\tilde{\theta}_{ev}$  over the three layers ( $n_{\perp}$  = number of layers) is calculated with the following equation. The thicknesses  $m_{\perp}$  of the layers are assumed to be 175, 150 and 275 mm respectively, hence, the topmost 60 cm are considered:

$$\tilde{\theta}_{ev} = \frac{\sum_{l=1}^{n_{t}} \frac{\Xi}{\theta_{ev,l}} \cdot m_{l}}{\sum_{l=1}^{n_{t}} m_{l}} \tag{4}$$

Finally, for all events the soil moisture storage change  $\Delta S$  [mm/ev] in the upper 60 cm is computed from the difference between =initial  $\frac{\Xi}{(\theta_{\text{ini},l} \text{[vol\%]})}$ and final soil moisture (Eq. 5, Table XX1).  $\frac{\Xi}{(\theta_{\text{fin},l} \text{[vol\%]})}$ that is converted to [vol/vol] via division by 100:

$$\Delta S = \sum_{l=1}^{n_{\rm f}} \frac{1}{100} \left( \overline{\overline{\overline{\theta}}}_{\rm fin,l} - \overline{\overline{\overline{\theta}}}_{\rm ini,l} \right) m_{\rm f} \tag{5}$$

For all plots and all events, the frequency distributions of the on alert soil moisture measurements are tested for normality with a Shapiro Wilk test at a significance level of  $\alpha = 0.05$ . It is also assessed whether significant differences between the four land use classes exist by performing a visual inspection of boxplots or histograms and Student t-tests. Moreover, standard deviations  $\sigma$  as measures of spatial variability at different scales are calculated at the plot scale ( $\sigma_j^{inner}$ , Eq. XX6, Table XX1) and at the catchment scale ( $\sigma_{cat.}^{inter}$ , Eq. XX7, Table XX1). Furthermore,  $\sigma$  is calculated between plots of the same land use ( $\sigma_{lu}^{inter}(t_{ev})$ , Eq. XX8, Table XX1) and between land use classes ( $\sigma^{betw}(t_{ev})$ , Eq. XX9, Table 1): the inner plot standard deviation  $\sigma_j^{inner}$ and the inter plot standard deviation for the land use classes grassland and vineyard ( $\sigma_{lu}^{inter}$ ;  $n_{j\in lu} > 1$ ) and the whole catchment ( $\sigma_{eat.}^{inter}$ ) are calculated for each event:

$$\frac{1}{\sigma_{j}^{\text{inner}}(t_{ev}) - \sqrt{\frac{1}{n_{i} - 1} \sum_{i=1}^{n_{i}} (\theta(x_{i,j}, t_{ev}) - \overline{\theta}_{j}(t_{ev}))^{2}}} \qquad (6)$$

$$\frac{\sigma_{iu}^{\text{inter}}(t_{ev}) - \sqrt{\frac{1}{n_{ju} - 1} \sum_{ju=1}^{n_{ju}} (\overline{\theta}_{j}(t_{ev}) - \overline{\theta}_{iu}(t_{ev}))^{2}}} \qquad j_{iu} \in lu; lu \in c_{iu} \qquad (7)$$

$$\frac{\sigma_{eut}^{\text{inter}}(t_{ev}) - \sqrt{\frac{1}{n_{jv} - 1} \sum_{j=1}^{n_{jv}} (\overline{\theta}_{j}(t_{ev}) - \overline{\theta}(t_{ev}))^{2}}} \qquad (8)$$

Furthermore, the between-land use standard deviation  $\sigma^{betw}$  is computed:

10

5
$$-\frac{1}{\sqrt{n_{e_{tw}}}} = \sqrt{\frac{1}{\frac{1}{n_{e_{tw}}}} - 1} \sum_{lw}^{n_{e_{tw}}} (\overline{\overline{\theta}}_{lw}(t_{ev}) - \overline{\overline{\theta}}_{ev})^2}$$
(9)

As an estimate of the uncertainty of the calculated plot and catchment mean values, the standard error of the plot\_-mean  $SEM_{j}^{inner}$  and the one of the catchment mean  $SEM_{cat.}^{inter}$  are calculated with Eq. (10) and Eq. (11). It should be noted that  $SEM_{cat.}^{inter}$  the latter is calculated from the on-alert measurements in the topsoil as well as from the continuous measurements over the soil profile,  $SEM_{j}^{inner}$  the former\_-only from the on-alert measurements. The SEM is used as a measure of the confidence that the sample mean corresponds to the universal mean; it increases with the standard deviation and decreases with the number of sampling points.

 $\frac{SEM_{j}^{\text{inner}}(t_{\text{ev}})}{SEM_{j}^{\text{inter}}(t_{\text{ev}})} - \frac{\sigma_{j}^{\text{betw}}(t_{\text{ev}})}{\sqrt{n_{j}}}$ (10)

Moreover, it is assessed whether temporal stability, i.e. consistency of soil moisture patterns at the catchment scale at different times of measuring (Vachaud et al., 1985), as reported by Huza et al. (2014) for six grassland plots in the Gazel catchment, is also found in the present on-alert data set: the relative spatial difference  $\delta_{j,ev}$  of each plot corresponds to the relative difference between the plot mean and the catchment mean (Eq. 12); its temporal mean  $\overline{\delta_j}$  is calculated with Eq. (213):

$$\delta_{j,ev} = \frac{\overline{\theta}_j(t_{ev}) - \overline{\overline{\theta}}_{ev}}{\overline{\overline{\theta}}_{ev}}$$
(12)

$$\overline{\delta_{j}} = \frac{1}{n_{ev}} \sum_{ev=1}^{n_{ev}} \delta_{j,ev}$$
(213)

The plot with the smallest  $\delta_{j,ev}$  is the one that agrees best with the catchment mean on a given time of measurement. The temporal variability of the spatial difference  $\sigma_{\delta_i}$  serves as an auxillary variable to assess whether this behavior is stable in time:

$$\sigma_{\delta_j} = \sqrt{\frac{1}{n_{\rm ev} - 1} \sum_{\rm ev=1}^{n_{\rm ev}} (\delta_{j,\rm ev} - \overline{\delta_j})^2}$$
(314)

5 The temporal dynamics of soil moisture during the autumn season and during single events that are captured with the continuous soil moisture measurements are analyzed and compared with the hyetographs and hydrographs. Furthermore, crosscorrelations are calculated for all continuously measuring sensors and all depths with the R function ccf (Gilbert and Plummer, 2016). For each plot the maxima of the crosscorrelation functions *L*CCmax between rainfall and the sensors in 10 cm depth, between the ones in 10 and 25 cm and the ones in 25 and 40 cm are calculated.

**10 **2.5 Hydrological response**

**2.5.1 Event based runoff coefficients**

In order to quantify the hydrological response of the catchment to different rainfall events, the dimensionless event-based runoff coefficient  $\phi_{ev}$  is calculated for all events:

$$\phi_{\rm ev} = \frac{Q_{\rm ev,cum}}{P_{\rm cum}} \tag{154}$$

- 15  $P_{eum}$  is calculated for each event as described in Sect. 2.3. To obtain cumulative event discharge  $Q_{ev,cum}$ , the time series of stream discharge  $Q_{tot}$  has to be separated into baseflow  $Q_b$  and event flow  $Q_{ev}$ .  $Q_{ev}$  is defined here to be the fast responding part of discharge that occurs during or directly after the rain event. It usually encompasses surface runoff or overland flow and fast subsurface flow.  $Q_b$  on the other hand is the slow responding part of discharge that lasts long after the rain event and feeds the stream between rain events. To obtain  $Q_{ev,cum}$ ,  $Q_{ev}$  is summed up over the whole period of the event. The onset of a discharge
- 20 event is defined as the first increase of discharge in response to a rain event. Defining the end of event discharge is more complicated and depends on the hydrograph separation method (Blume et al., 2007). Usually, the end of event flow is defined as the moment when  $Q_b$  equals  $Q_{tot}$ , but for some events the onset of a second event impedes this procedure which causes errors. Taking into consideration that there is no standard method for hydrograph separation and that results obtained with different methods can differ substantially (Blume et al., 2007), seven different hydrograph separation techniques that are
- 25 described in the following section are applied and compared (Sect. 2.5.2) (Fig. 2). The uncertainty of  $Q_{tot}$  associated with the

stage-discharge relation can be important especially for high-flow conditions. This was taken into account by calculating  $\phi_{ev}$  with the upper and the lower limit of the 90 % confidence interval of discharge obtained with the BaRatin framework. Because several of the assumptions underlying standard regression analysis methods (normal distribution of dependent and

independent variables, only the dependent variable is subject to error, homoscedasticity) are not met, it is not attempted to set-

5 up linear or non linear regression models that predict  $\phi_{ev}$  from possible explanatory variables such as initial soil moisture, initial baseflow, rainfall depth or rain intensity. The relation of these variables and  $\phi_{ev}$  is considered nonetheless.

**2.5.2 Hydrograph separation**

**Straight line method.** The straight line method (SL) is a simple, graphical method where baseflow during the event is interpolated by connecting the point in the event hydrograph at which discharge first increases as a response to the rain event

10 with the first point on the falling limb of the hydrograph with the same discharge value. As this condition is often never met, the end of event flow is often determined by the onset of the next event or discharge below a threshold.

**Constant-k method.** The CK-method proposed by Blume et al. (2007) is based on the assumption that baseflow recession behaves similar to the outflow of a linear storage. Thus, baseflow at time step *t* can be described as exponential recession with the recession parameter *k* and initial flow  $Q_0$ :

$$Q_{\rm b}(t) = Q_0 \,{\rm e}^{-k\,t} \tag{165}$$

k is calculated at each time step by differentiating eq.  $\frac{16-5}{2}$  and division by  $Q_{b}(t)$ :

$$k = -\frac{\mathrm{d}Q}{\mathrm{d}t} \frac{1}{Q_{\mathrm{b}}(t)} \tag{176}$$

Event flow is assumed to terminate at time step  $t_e$  which is defined as the end of event runoff, once k becomes approximately constant. Baseflow is assumed to be equal to the discharge before the onset of event flow up to  $t_e$  when it equals  $Q_{tot}$ .

20 Electrical conductivity method. Hydrograph separation is also conducted based on electrical conductivity (*EC*) which serves as a natural tracer (Miller et al., 2014; Pellerin et al., 2008). The method relies on the assumption that stream flow  $Q_{tot}$  with electrical conductivity  $EC_{tot}$  is a mixture of subsurface flow  $Q_{sb}$  and surface flow  $Q_s$ , which have significantly different *EC* signals  $EC_{sb}$  and  $EC_s$  (Nakamura, 1971):

$$Q_{\rm tot} = Q_{\rm sb} + Q_{\rm s} \tag{187}$$

$$Q_{\rm tot} \cdot EC_{\rm tot} = Q_{\rm sb} \cdot EC_{\rm sb} + Q_{\rm s} \cdot EC_{\rm 
[revised manuscript text omitted]

15